# Automated extraction of chemical synthesis actions from experimental procedures

Alain C. Vaucher [1,2 ✉], Federico Zipoli[1,2], Joppe Geluykens [1], Vishnu H. Nair[1], Philippe Schwaller [1] & Teodoro Laino[1]

Experimental procedures for chemical synthesis are commonly reported in prose in patents or in the scientific literature. The extraction of the details necessary to reproduce and validate a synthesis in a chemical laboratory is often a tedious task requiring extensive human intervention. We present a method to convert unstructured experimental procedures written in English to structured synthetic steps (action sequences) reflecting all the operations needed to successfully conduct the corresponding chemical reactions. To achieve this, we design a set of synthesis actions with predefined properties and a deep-learning sequence to sequence model based on the transformer architecture to convert experimental procedures to action sequences. The model is pretrained on vast amounts of data generated automatically with a custom rule-based natural language processing approach and refined on manually annotated samples. Predictions on our test set result in a perfect (100%) match of the action sequence for 60.8% of sentences, a 90% match for 71.3% of sentences, and a 75% match for 82.4% of sentences.

[1] IBM Research Europe, Säumerstrasse 4, Rüschlikon, 8803 Switzerland. [2]These authors contributed equally. Alain C. Vaucher, Federico Zipoli. ✉email: ava@zurich.ibm.com

In chemistry like in other scientific disciplines, we are witnessing the growth of an incredible amount of digital data, leading to a vast corpus of unstructured media content—including articles, books, images and videos—rarely with any descriptive metadata. While scientists have developed several technologies for analyzing and interacting with unstructured data, quite often these solutions rely on identifying and utilizing rules specific to each data item at the cost of a substantial human effort. Currently, the processing of unstructured data is pivotal to the work of many scientists: it transforms this data into a structured form that is easily searchable and that can be combined easily with automated workflows.

The availability of structured chemical data is especially important for automation due to the increasing interest in robots in the context of organic synthesis[1–4]. Structured data is also important to stimulate the design of predictive models for optimizing reaction procedures and conditions, similar to the success of the AI-guided reaction prediction schemes[5–8] for organic molecules.

In fact, although some simple organic reaction data are widely presented in well-structured and machine readable format, this is not the case for the corresponding chemical reaction procedures which are reported in prose in patents and in scientific literature. Therefore, it is not surprising if their conversion into a structured format is still a daunting task. As a consequence, the design of an automated conversion from unstructured chemical recipes for organic synthesis into structured ones is a desirable and needed technology.

Ultimately, with such an algorithm, a machine could ingest an experimental procedure and automatically start the synthesis in the lab, provided that all the necessary chemicals are available. Also, if applied to a large collection of experimental procedures, the conversion to structured synthesis actions could prove interesting for the analysis of reaction data, and could facilitate the discovery of patterns and the training of machine-learning models for new organic chemistry applications.

In this work, we focus on the conversion of experimental procedures into series of structured actions, with an emphasis on organic chemistry. To do so, we first identify general synthesis tasks covering most of the operations traditionally carried out by organic chemists. We implement and discuss several computational approaches for the extraction of such structured actions from experimental procedures. Rule-based models represent a good starting point for this endeavor, but they are quite sensitive to the formulation of the rules and to noise in the experimental procedures, such as typing errors or grammar mistakes[3]. We therefore introduce a deep-learning model based on the transformer architecture to translate experimental procedures into synthesis actions. We pretrain it on data generated with rule-based models and refine it with manually annotated data.

In doing so, our goal is for the sequence of actions to correspond to the original experimental procedure as closely as possible, with all the irrelevant information discarded. This means that an extracted action sequence contains, in principle, all the details required by a bench chemist or a robotic system to conduct a reaction successfully.

Retrieving information from the chemistry literature has received a lot of attention over the last decades[9,10]. One of the predominant goals is to mine information from patents, papers and theses, and save it as structured data in databases in order to make chemical knowledge searchable and enable queries about materials or properties. Due to the complex syntax of chemical language, a lot of effort has been put into the development of named entity recognition methods for chemistry. Named entity recognition entails the automatic detection of relevant words or word groups in a text and their assignment in categories. Typical approaches apply rules and dictionaries, machine-learning, or combinations thereof[9]. For instance, many named entity recognition methods have been applied to the detection of chemical entities (compound names and formulas) in text (see, for instance, refs. [11–15], as well as ref. [9] for an extensive review).

Other approaches apply named entity recognition to also detect other chemistry-related information such as operations or reaction conditions. The ChemicalTagger tool, which focuses on the experimental sections of scientific text, parses different kinds of entities and determines the relationships between them[16]. Thereby, it also identifies so-called action phrases that associate text excerpts to actions. ChemDataExtractor aims to extract as much data as possible from the scientific literature to populate chemical databases[17]. It does not focus solely on experimental procedures and is also able to extract spectroscopic attributes or information present in tables, for instance. Weston et al. follow a similar strategy and apply their method on materials science abstracts with the goal to produce easily searchable knowledge databases[18].

In the field of materials science, several text-mining tools have been applied to the study of synthesis procedures. Kim et al. designed a pipeline for the extraction of synthesis parameters which allows them to examine and compare synthesis conditions and materials properties across many publications[19,20]. In another work, they applied this pipeline to extract synthesis data for specific materials and train a variational autoencoder that generates potential synthesis parameter sets[21]. More recently, data extracted with the same tools allowed machine-learning models to learn to predict the precursors and sequence of actions to synthesize inorganic materials[22]. Mysore et al. applied text-mining tools to convert synthesis procedures to action graphs[23]. The nodes of the action graphs represent compounds, actions, or experimental conditions, and they are connected by edges that represent the associations between the nodes. Huo et al. applied latent Dirichlet allocation to cluster sentences of experimental procedures into topics in an unsupervised fashion, and then designed a machine-learning model to classify documents into three synthesis categories based on their topic distribution[24]. In an effort to facilitate the design and training of future machine-learning models, Mysore et al. provided a dataset of 230 annotated materials synthesis procedures[25]. A similar effort had been presented earlier for web lab protocols in biology[26].

The extraction of synthesis information for organic chemistry has received less attention. Recently, Cronin and co-workers developed a robotic system able to perform organic synthesis autonomously[3], requiring a synthetic scheme described in the so-called chemical descriptive language (XDL). They implement a rudimentary tool for translating a given procedure into XDL that follows the identification of key entities in the text and assembling the corresponding list of operations, using existing natural language-processing tools. This approach is exposed to linguistic challenges and its success depends to a large extent on how the experimental procedure is formulated. As a consequence, creating the XDL schemes remains largely manual. The Reaxys[27] and SciFinder[28] databases are also worth mentioning in the context of extracted organic synthesis information. These commercial databases contain reaction data (such as reagents, solvents, catalysts, temperatures, and reaction duration) for a large number of chemical reactions. These data are usually extracted from the scientific literature and curated by expert scientists.

To contrast the present work from previous approaches, our model converts experimental procedures as a whole into a structured, automation-friendly format, instead of scanning texts in search of relevant pieces of information. We aim for this conversion to be as reliable as possible, with the goal to make human verification unnecessary. Also, in contrast to other

approaches, our deep-learning model does not rely on the identification of individual entities in sentences. In particular, it does not require specifying which words or word groups the synthesis actions correspond to, which makes the model more flexible and purely data-driven.

The trained deep-learning model for the extraction of action sequences is available free of charge on the cloud-based IBM RXN for Chemistry platform[29].

## Results

**Synthesis actions**. The experimental procedures we consider in this work come from patents and represent single reaction steps. To conduct the full synthesis of a molecule, several such reaction steps are combined. The following is an example of a typical experimental procedure that is to be converted to automation-friendly instructions (which will be given further below in Table 2):

To a suspension of methyl 3-7-amino-2-[(2,4-dichlorophenyl)(hydroxy)methyl]-1H-benzimidazol-1-ylpropanoate (6.00 g, 14.7 mmol) and acetic acid (7.4 mL) in methanol (147 mL) was added acetaldehyde (4.95 mL, 88.2 mmol) at 0 °C. After 30 min, sodium acetoxyborohydride (18.7 g, 88.2 mmol) was added. After 2 h, the reaction mixture was quenched with water, concentrated in vacuo, diluted with ethyl acetate, washed with aqueous sodium hydroxide (1 M) and brine, dried over sodium sulfate, filtered and concentrated in vacuo. The residue was purified by column chromatography on silica gel eluting with a 10–30% ethyl acetate/n-hexane gradient mixture to give the title compound as a colorless amorphous (6.30 g, 13.6 mmol, 92%).

From such an experimental procedure, our goal is to extract all relevant information to reproduce the chemical reaction, including details about work-up. The structured format into which we convert this information consists of a sequence of synthesis actions. It is to be noted that restricting syntheses to the sequential execution of actions prevents us from supporting non-linear workflows. However, such branched synthesis procedures are rare when considering single reaction steps (see "Discussion" section). Furthermore, they can partly be remedied by the choice of actions, as will be explained below.

The predefined set of synthesis actions must be flexible enough to capture all the information necessary to conduct the chemical reactions described in experimental procedures. We tailored our set of actions to best reflect the content of experimental procedures as commonly described in patents. Accordingly, our actions cover operations of conventional batch chemistry for organic synthesis. We note that synthesis actions have been defined as well in other work. For instance, Hawizy et al. define a set of 21 types of so-called action phrases for experimental procedures from patents[16]. In the context of materials science, Huo et al. interpret topics extracted by a latent Dirichlet allocation as categories of experimental steps[24], and Kim et al. cluster actions into a set of 50 categories in an automated procedure[22].

The actions we selected are listed in Table 1. Each action type has a set of allowed properties. For instance, the Stir action can be further specified by a duration, a temperature, and/or an atmosphere (and nothing else). The properties allowed for each action type are listed and explained in the Supplementary Note 1 and Supplementary Table 1.

Most action types listed in Table 1 correspond to actual synthesis operations with direct equivalents in the wet laboratory. We note that drying and washing, in organic synthesis, correspond to different operations depending on their context. In particular, the additional properties attached to the two types of drying are different and we therefore define two action types

for drying, DrySolid and DrySolution. MakeSolution describes the preparation of a separate solution. This enables us to support experimental procedures that require solutions or mixtures to be prepared separately for use in another action. Accordingly, MakeSolution is important in ensuring the compatibility with a linear sequence of actions, by avoiding the necessity to consider multiple reactors in an action sequence. We ignore information about glassware and apparatus on purpose, as this is largely imposed by the availability of equipment or the scale of the reaction, and the reaction success should not depend on it.

A few action types do not actually correspond to laboratory operations, but are convenient when retrieving information from experimental procedures. The FollowOtherProcedure action type is selected when the text refers to procedures described elsewhere, in which case no actual actions can be extracted. NoAction is assigned to text that does not relate to a synthesis operation, such as nuclear magnetic resonance data or sentences describing the physical properties of the reaction mixture. The OtherLanguage action covers experimental procedures that are not written in English. InvalidAction indicates that a text fragment is relevant but cannot be converted to one of the actions defined above. This action type is for instance selected for synthesis operations that are not covered by the actions of Table 1, or for branched synthesis procedures.

When determining the actions corresponding to an experimental procedure, it is important to consider that some actions are implicit. For instance, in the sentence "The organic layer was dried over sodium sulfate", the phase separation and collection of the organic layer is implicit (no verb) and will result in a CollectLayer action preceding DrySolution. Similarly, "23 g of aluminum chloride in 30 mL of dichloroethane was heated to 50 °C." corresponds to three actions (MakeSolution, Add, SetTemperature) although the sentence contains only one verb ("heat").

A single action type may cover a wide range of formulations present in experimental procedures. For instance, an Add action can be expressed using the English verbs "add", "combine", "suspend", "charge", "dilute", "dissolve", "mix", "place", "pour", and "treat", among others. As an additional example, a Concentrate action can be described in terms of concentrating a solution, evaporating a solvent, as well as removing a solvent or distilling it off.

Furthermore, an English verb may correspond to different actions depending on its context. For instance, "heat" may, on the one hand, indicate a punctual change in temperature for subsequent actions, or, on the other hand, inform that the reaction mixture should be heated for a specified duration. In the former case, we convert it to a SetTemperature action, and in the latter case to a Stir action. Another example is the verb "remove", which may relate to Concentrate when associated with a solvent or to Filter in the context of a filtration.

It is important to consider that there can be multiple ways to assign actions to some synthesis operations. For example, the Quench and PH actions can, in principle, both be formulated as Add actions. Also, a Partition action can be expressed as two Add actions followed by a PhaseSeparation action. In such cases, we want to preserve the intent of the original experimental procedure and keep the variant closest to the text. We also note that the action scheme not only supports experimental procedures written in terms of specific reagents, but also the ones referring to general reagents (for instance, "the aldehyde" instead of "4-hydroxy-3-methoxybenzaldehyde").

Computationally, actions can be stored as items associating the action type with a set of properties (complying with the available properties for each action type). For practical purposes, we define

**Table 1 Action types for information extraction from experimental procedures.**

| Action name | Description |
|---|---|
| Add | Add a substance to the reactor |
| CollectLayer | Select aqueous or organic fraction(s) |
| Concentrate | Evaporate the solvent (rotavap) |
| Degas | Purge the reaction mixture with a gas |
| DrySolid | Dry a solid |
| DrySolution | Dry an organic solution with a desiccant |
| Extract | Transfer compound into a different solvent |
| Filter | Separate solid and liquid phases |
| MakeSolution | Mix several substances to generate a mixture or solution |
| Microwave | Heat the reaction mixture in a microwave apparatus |
| Partition | Add two immiscible solvents for subsequent phase separation |
| PH | Change the pH of the reaction mixture |
| PhaseSeparation | Separate the aqueous and organic phases |
| Purify | Purification (chromatography) |
| Quench | Stop reaction by adding a substance |
| Recrystallize | Recrystallize a solid from a solvent or mixture of solvents |
| Reflux | Reflux the reaction mixture |
| SetTemperature | Change the temperature of the reaction mixture |
| Sonicate | Agitate the solution with sound waves |
| Stir | Stir the reaction mixture for a specified duration |
| Triturate | Triturate the residue |
| Wait | Leave the reaction mixture to stand for a specified duration |
| Wash | Wash (after filtration, or with immiscible solvent) |
| Yield | Phony action, indicates the product of a reaction |
| FollowOtherProcedure | The text refers to a procedure described elsewhere |
| InvalidAction | Unknown or unsupported action |
| OtherLanguage | The text is not written in English |
| NoAction | The text does not correspond to an actual action |

a bijective conversion to and from a textual representation of the actions. This textual representation is concise and easily understandable. It contains, for each action, all the non-empty properties of that action. With that format, the textual representation of the actions corresponding to the experimental procedure quoted above is shown in Table 2.

**Models for action sequence extraction.** We studied several models for the automated extraction of action sequences from experimental procedures available in the Pistachio dataset[30].

A first possibility is to parse the text for information about operations, compounds, quantities, and other conditions. This can be achieved by inspecting the structure of the sentences in the experimental procedures to detect the relevant pieces of information with the help of rules. In this work, we look into two such rule-based methods (see "Methods" section for details). These models require meticulous work when formulating extraction rules. Still, they do not always lead to an ideal conversion of experimental procedures into action sequences: it is virtually impossible to define rules covering every possible way to describe a synthesis, while at the same time being robust to noise in the experimental procedures.

To improve the quality of the extracted actions, we also look into machine learning for this task. As machine-learning models learn from data instead of rules, they are more flexible than rule-based models, which usually results in a greater robustness to noise. In our case, the training data can even be provided by the rule-based models in an initial phase. Concretely, we combine the action sequences generated by rule-based approaches into a pretraining dataset used for the initial training of the machine-learning model. We then refine the pretrained model with manually annotated samples of higher quality. To achieve this, we design a deep-learning model relying on a transformer-based

**Table 2 Action sequence extracted from an experimental procedure.**

| | |
|---|---|
| 1 | MakeSolution with methyl 3-7-amino-2-[(2,4-dichlorophenyl)(hydroxy)methyl]-1H-benzimidazol-1-ylpropanoate (6.00 g, 14.7 mmol) and acetic acid (7.4 mL) and methanol (147 mL); |
| 2 | Add SLN; |
| 3 | Add acetaldehyde (4.95 mL, 88.2 mmol) at 0 °C; |
| 4 | Wait 30 min; |
| 5 | Add sodium acetoxyborohydride (18.7 g, 88.2 mmol); |
| 6 | Wait 2 h; |
| 7 | Quench with water; |
| 8 | Concentrate; |
| 9 | Add ethyl acetate; |
| 10 | Wash with aqueous sodium hydroxide (1 M); |
| 11 | Wash with brine; |
| 12 | DrySolution over sodium sulfate; |
| 13 | Filter keep filtrate; |
| 14 | Concentrate; |
| 15 | Purify; |
| 16 | Yield title compound (6.30 g, 13.6 mmol, 92%). |

The sequence corresponds to the example experimental procedure given above.

encoder–decoder architecture that defines the extraction task as a translation of experimental procedure text into the textual representation of the associated actions.

In order to improve the performance of the refined machine-learning model, we perform additional refinement experiments involving data augmentation of the annotated samples. We also evaluate ensembles of trained models and, for comparison purposes, we train another model on the annotation dataset only (i.e. without pretraining).

The source of the experimental procedure data and all the above-mentioned approaches for action sequence extraction are detailed in the "Methods" section.

**Model evaluation**. We evaluate all the approaches on the test set of the annotation dataset. This set is made up of sentences that are more complex than the average, since the sentences selected for annotation represent cases that the rule-based models struggled with (see the "Methods" section).

In Table 3, we show six metrics to compare different models for action sequence extraction. For clarity and conciseness, this table lists a selection of models only. Details related to this selection, as well as a comparison of all the refinement experiments, can be found in the Supplementary Note 2. The validity is a measure of syntactical correctness of the textual representation of actions. It is given as the fraction of predictions that can be converted back to actions (as defined in Table 1) without error. The BLEU score[31] is a metric commonly used to evaluate models for machine translation. We adapted its calculation in order not to penalize predictions containing less than four words (see the Supplementary Note 3 for details). The Levenshtein similarity is calculated by deducting the normalized Levenshtein distance[32] from one, as implemented in the `textdistance` library[33]. The 100%, 90%, and 75% accuracies are the fractions of sentences that have a normalized Levenshtein similarity of 100%, 90%, 75% or greater, respectively. Accordingly, the 100% accuracy corresponds to the fraction of sentences for which the full action sequence is predicted correctly, including the associated properties.

As expected, the combined rule-based model and the deep-learning model pretrained on the rule-based data have a similar performance. Upon inspection, it appears that the better metrics of the deep-learning variant can be explained by sentences that the rule-based model classified as `InvalidAction` and that the pretrained model was partially able to predict correctly. Training a model on the annotated data only (no pretraining) leads to a model with a better accuracy than the one relying on pretraining only. Refining the pretrained translation model results in a considerable improvement compared to the other models. It more than doubles the fraction of sentences that are converted correctly compared to the pretrained model. Refining the model, however, slightly decreases the action string validity. The corresponding invalid predictions are converted to `InvalidAction`. Also, Table 3 illustrates that omitting the pretraining step leads to a considerably lower model accuracy. In the following, we only consider the refined translation model for analysis and discussion.

Inspection of the actions extracted by this model provides interesting insight into its strengths and weaknesses. For the incorrectly predicted action sequences, the differences are often limited to a single action. In some cases, it is even ambiguous which of the prediction or the ground truth (hand annotation) is better. In other cases, however, the predictions are clearly incorrect. Table 4 shows the ground truth and the predicted action sequences for a selection of sentences. In the Supplementary Data 1, the interested reader may find, as additional examples, all the experimental procedure sentences from the annotation test set with the corresponding actions extracted by the different models.

---

**Table 3 Metrics for the extraction of synthesis actions.**

| Model | Validity | BLEU score | Levenshtein similarity | 100% accuracy | 90% accuracy | 75% accuracy |
|---|---|---|---|---|---|---|
| Combined rule-based model | **100.0** | 51.5 | 60.1 | 21.9 | 29.0 | 42.6 |
| Pretrained translation model | **100.0** | 58.6 | 68.7 | 24.7 | 33.2 | 48.3 |
| Model without pretraining | 98.9 | 64.7 | 76.4 | 37.8 | 47.7 | 62.8 |
| Refined translation model | 99.4 | **85.0** | **86.9** | **60.8** | **71.3** | **82.4** |

The metrics are evaluated on the annotation test set for the approaches introduced in this work. All values are given in %, and the best values are indicated in bold. An extended table showing the metrics for all the refinement experiments can be found in the Supplementary Note 2.

---

**Table 4 Example of extracted action sequences.**

*After adjusting to pH 1.5 with 10% hydrochloric acid, the ethyl acetate solution was separated, washed with a saturated aqueous sodium chloride and then dried over anhydrous magnesium sulfate.*
(1) PH with 10% hydrochloric acid to pH 1.5; PHASESEPARATION; COLLECTLAYER organic; WASH with saturated aqueous sodium chloride; DRYSOLUTION over anhydrous magnesium sulfate.
(2) PH with 10% hydrochloric acid to pH 1.5; PHASESEPARATION; COLLECTLAYER organic; WASH with saturated aqueous sodium chloride; DRYSOLUTION over anhydrous magnesium sulfate.

*A solution of solid sodium metal (450 mg, 19.75 mmol) in EtOH at 30 °C. was treated with ethyl acetoacetate (103 g, 790 mmol) maintaining temperature.*
(1) MAKESOLUTION with sodium metal (450 mg, 19.75 mmol) and EtOH; ADD SLN; ADD ethyl acetoacetate (103 g, 790 mmol) at 30 °C.
(2) MAKESOLUTION with **solid** sodium metal (450 mg, 19.75 mmol) and EtOH; ADD SLN; ADD ethyl acetoacetate (103 g, 790 mmol) at 30 °C.

*3-Bromo-2-fluoroaniline (10 g, 52.63 mmol) was dissolved in DCM (100 mL) under nitrogen atmosphere.*
(1) ADD 3-Bromo-2-fluoroaniline (10 g, 52.63 mmol); ADD DCM (100 mL) under nitrogen.
(2) ADD 3-Bromo-2-fluoroaniline (10 g, 52.63 mmol) **under nitrogen;** ADD DCM (100 mL) under nitrogen.

*Upon complete addition, the reaction mixture was allowed to warm to room temperature and the reaction was stirred for about 12 h.*
(1) STIR for 12 h at room temperature.
(2) **SETTEMPERATURE room temperature;** STIR for **about** 12 h.

*The residue was crystallized from 60 ml of benzotrifluoride, during this operation, the mixture was briefly boiled with activated carbon and filtered whilst still hot.*
(1) INVALIDACTION.
(2) **RECRYSTALLIZE from benzotrifluoride (60 ml); FILTER keep precipitate**.

For sentences picked from experimental procedures, the actions sequences predicted by the refined translation model (2) are compared to the annotated sequences (1). The errors in the prediction are highlighted in bold. The action sequences predicted by the other models, as well as predictions on other sentences, can be found in the Supplementary Data 1.

**Table 5 Prediction accuracy by action type.**

| Action type | Type match | Full match | Only in prediction | Only in ground truth |
|---|---|---|---|---|
| Add | 246 | 185 | 21 | 9 |
| Stir | 112 | 100 | 2 | 6 |
| MakeSolution | 57 | 46 | 5 | 5 |
| SetTemperature | 55 | 52 | 6 | 5 |
| Concentrate | 48 | 48 | 3 | 6 |
| Wash | 44 | 43 | 3 | 1 |
| PH | 41 | 34 | 2 | 2 |
| CollectLayer | 35 | 35 | 4 | 2 |
| Extract | 32 | 31 | 0 | 2 |
| Filter | 32 | 29 | 4 | 2 |
| Yield | 31 | 25 | 5 | 6 |
| NoAction | 22 | 22 | 3 | 3 |
| DrySolution | 22 | 21 | 2 | 0 |
| Purify | 19 | 19 | 2 | 5 |
| Wait | 16 | 15 | 3 | 3 |
| FollowOtherProcedure | 14 | 14 | 3 | 1 |
| DrySolid | 10 | 9 | 2 | 2 |
| Quench | 7 | 7 | 0 | 1 |
| Reflux | 7 | 5 | 0 | 0 |
| Partition | 5 | 4 | 0 | 0 |
| PhaseSeparation | 4 | 4 | 0 | 0 |
| Triturate | 3 | 2 | 2 | 0 |
| OtherLanguage | 2 | 2 | 0 | 0 |
| Recrystallize | 2 | 0 | 2 | 0 |
| Degas | 1 | 1 | 1 | 0 |
| InvalidAction | 0 | 0 | 5 | 11 |

The table indicates the number of actions for which the type was predicted correctly (type match), the number of actions for which not only the type, but also the associated properties, were predicted correctly (full match), the number of actions of a given type that were present only in the prediction, and the number of actions of a given type that were present only in the ground truth.

In Table 5, we show the accuracy of the predictions on the annotation test set by action type. It illustrates that for most actions, not only the type but also the associated properties are predicted correctly. Interestingly, no InvalidAction of the ground truth is present in the predictions, and multiple InvalidAction actions are predicted when the original sentence is not invalid. This problem is difficult to alleviate, since InvalidActions in the annotations often correspond to unusual and infrequent operations or formulations.

Figure 1 illustrates, for the actions present in the ground truth, the corresponding action types predicted by the transformer model. Most of the incorrectly predicted actions relate to NoAction, InvalidAction, or actions with no counterpart. Other than that, very few actions are predicted incorrectly. Interesting errors are mixing up MakeSolution and Add (three times), predicting DrySolution instead of DrySolid (two times) and Wait instead of Stir (two times), or a PH action that is considered to be an Add action. More insight into the incorrect predictions can be gained by looking into the Supplementary Data 1 mentioned earlier.

To better understand the errors of the model, we also take advantage of the ability of the model to make multiple suggestions for translation with a beam search. This is especially interesting for the sentences that the model is least confident about. The five best action sequences suggested by the refined model for all the sentences in the annotation test set can be found in the Supplementary Data 2.

**Data insights**. Visualization of the extracted actions gives us interesting insight into the chemistry described in patents, and into the models presented in this work.

First, Fig. 2a, b displays the distribution of the number of characters and the number of actions for sentences from Pistachio (used for pretraining) and from the annotation dataset. The left figure shows that both sentence length distributions are similar, and are characterized by an average sentence length of around 100 characters. The annotation dataset contains fewer very short and fewer very long sentences. The right figure shows that most sentences (roughly one-third) describes one single action, with a decreasing probability to find sentences with increasingly many actions. The differences between both distributions can be explained by differences in the underlying sentences (Pistachio vs. annotation dataset) and by the different extraction approach (rule-based model vs. hand annotations).

Figure 2c shows the distribution of actions extracted by the rule-based model on the Pistachio dataset and on the annotation dataset. As a whole, both distributions are similar, and they give an idea of the frequency of chemical operations in patents. One can for instance observe that addition, stirring and concentration belong to the most common operations, while only few experimental procedures involve recrystallization, microwaving or sonication. The differences between both distributions reflect the criteria for the selection of the sentences to annotate. For instance, the rule-based model tags too many sentences as InvalidAction, and therefore it is sensible to annotate as many such sentences as possible. Further below, Fig. 3 will show that the rule-based model overestimates the frequency of InvalidActions. One can also see that PH actions are overrepresented in the annotations, because of the necessity to parse the pH value and the current inability of the rule-based model to do so.

In Fig. 2d, one can see the distribution of hand-annotated actions on the full annotation set of 1764 samples and on its subset from the test split containing 352 samples. This figure shows that the distribution of actions in the test split is close to the one of the full annotation set, and hints that it catches sufficient diversity for evaluating the models studied in this work.

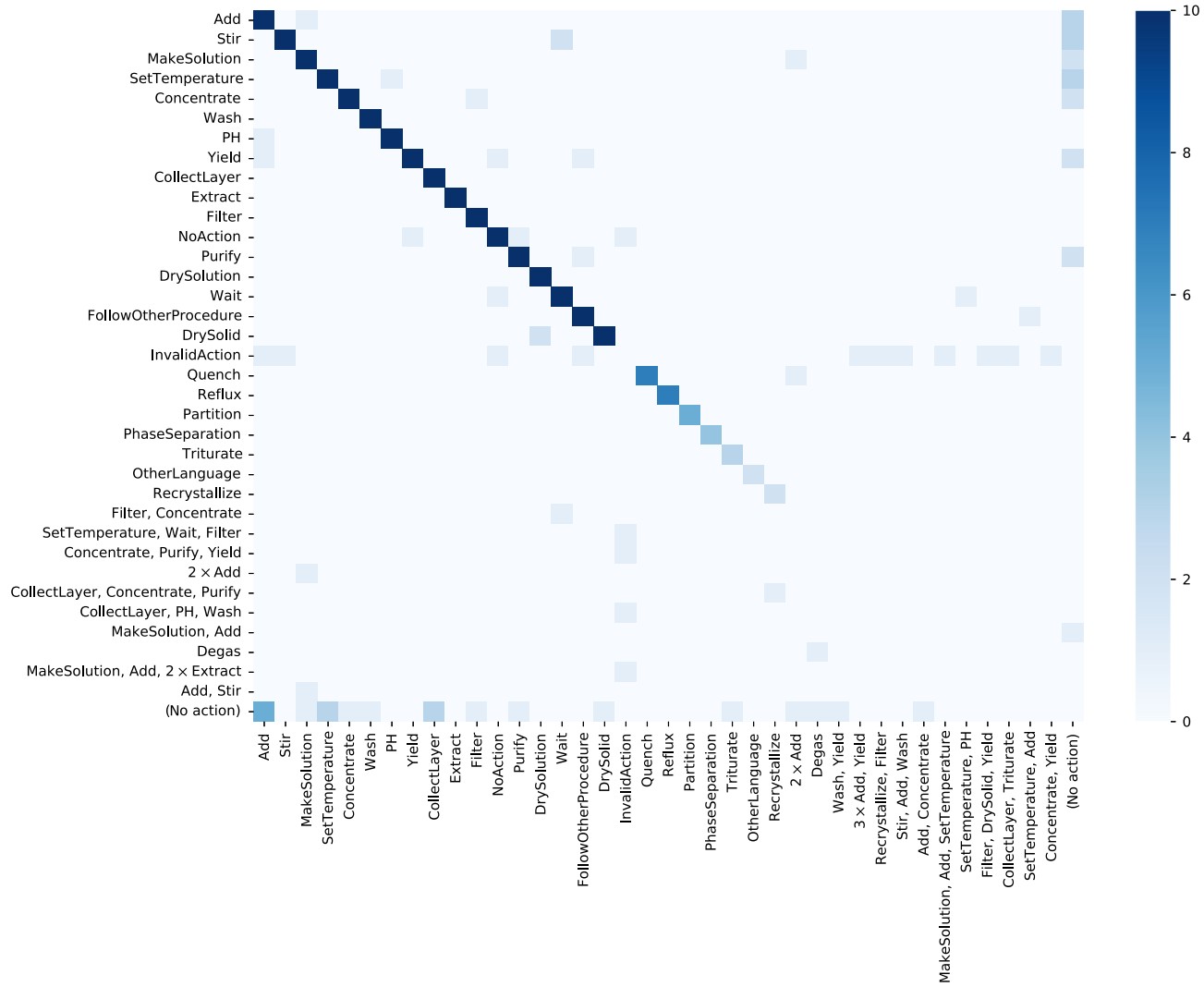

**Fig. 1 Visualization of the correctness of predicted action types.** The action types predicted by the transformer model (labels on the *x*-axis) are compared to the actual action types of the ground truth (labels on the *y*-axis). This figure is generated by first counting all the correctly predicted action types (values on the diagonal); these values correspond to the column "Type match'' of Table 5. Then, the off-diagonal elements are determined from the remaining (incorrectly predicted) actions. Thereby, the last row and column gather actions that are present only in the predicted set or ground truth, respectively. For clarity, the color scale stops at 10, although many elements (especially on the diagonal) exceed this value.

Figure 3 illustrates the actions predicted by the rule-based and machine-learning models on the annotation test set, compared with the hand-annotated actions. One can see that the distribution of actions predicted by the machine-learning model follows very closely the ground truth distribution. In particular, the frequency of NoAction and InvalidAction is much closer to the ground truth than the rule-based model, although the frequency of InvalidAction is underestimated.

## Discussion
The present work demonstrates the ability of a transformer-based sequence-to-sequence model to extract actions from experimental procedures written in prose. Training such a model on automatically generated data is already sufficient to achieve a similar accuracy as the rule-based approaches that produced that data. Enhancing the training data with manually annotated samples rapidly shows the advantage of a data-driven approach, since a relatively small set of annotations already leads to a dramatic improvement in accuracy. The ability of the model to learn a complex syntax with a different set of properties for each action type avoids the necessity to design a complex deep-learning

model taking into account multiple output types and demonstrates the power of the transformer architecture.

This work represents an important first step towards the automatic execution of arbitrary reactions with robotic systems. Before this is possible, however, it will be necessary to develop methods to infer information missing from experimental procedures. For instance, experimental procedures sometimes do not specify the solvents used for some operations, their quantities, or operation durations.

While the actions defined in this work are able to cover a large majority of experimental procedures, we are aware of some shortcomings of our approach. The choice to only support linear sequences of actions prevents us from addressing cross-references over long distances in the text. The MakeSolution and CollectLayer partly alleviate this disadvantage by encapsulating the preparation of a solution taking place in a separate flask, and by allowing for combining multiple solvent fractions generated during work-up, respectively. Then, in our annotation dataset of 1764 sentences, only four sentences correspond to an unsupported nonlinear sequence of actions. They are given as an illustration in the Supplementary Note 4. Other than that, the

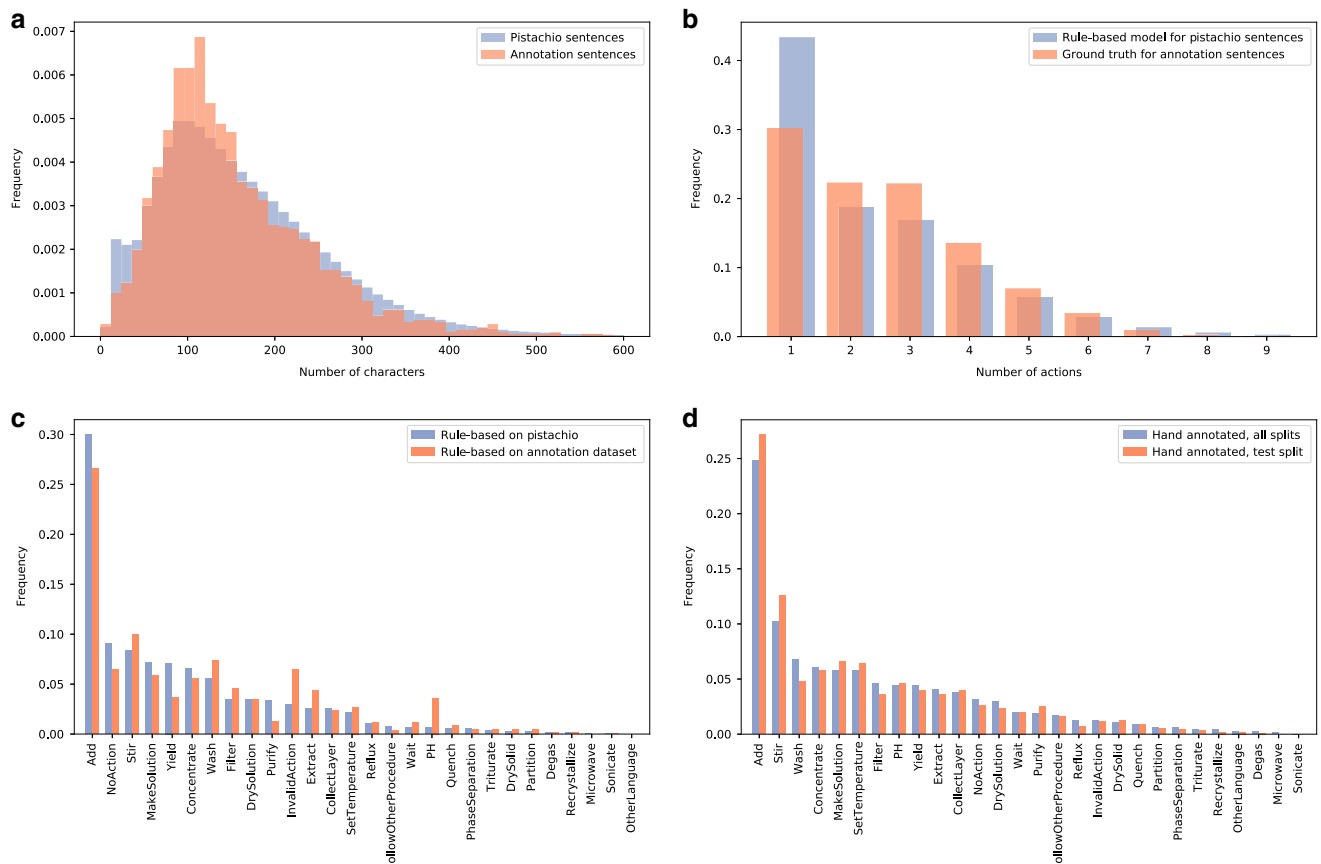

**Fig. 2 Statistics of the Pistachio and annotation datasets. a** Distribution of the number of characters for sentences from Pistachio and from the annotation dataset. **b** Distribution of the number of actions per sentence. For the Pistachio dataset, this number is computed from the actions extracted by the rule-based model. For sentences from the annotation dataset, this number is determined from the ground truth (hand annotations). **c** Distribution of action types extracted by the rule-based model on the Pistachio dataset and on the annotated dataset. The action types are ordered by decreasing frequency for the Pistachio dataset. **d** Distribution of action types determined from hand annotations for the full annotation dataset and its test split. The action types are ordered by decreasing frequency for the full annotation dataset.

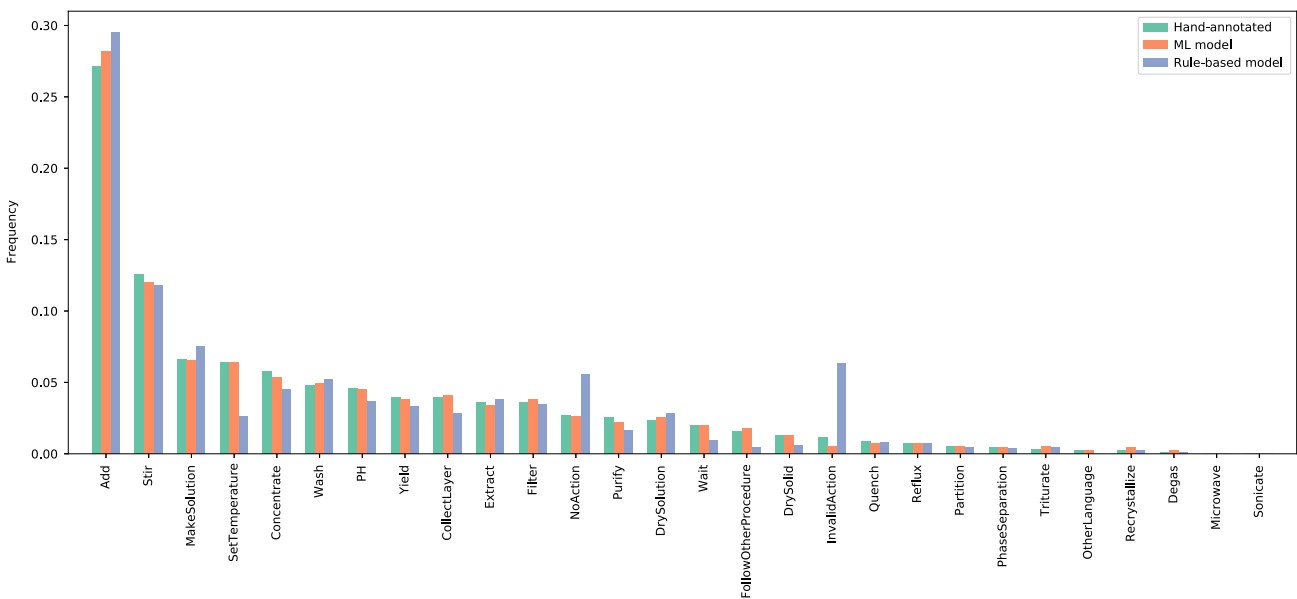

**Fig. 3 Distribution of action types of the annotation test set.** The action types are ordered by decreasing frequency for the hand annotations.

current format does not allow operations that depend on the state of the system. In particular, formulations indicating until when an operation must be performed ("until the color disappears", "until half the solvent has evaporated", and so on) are usually not specific enough to be supported by our action definitions.

Another limitation originates in our specific choice of action types (Table 1) and corresponding properties, which does not yet allow for a 100% coverage of the operations in organic chemistry. This limitation can be alleviated by extending the action definitions, which is a process guided mainly by time and experience. In the Supplementary Note 5, we give a few examples of such limitations, as well as suggestions for addressing them.

The rule-based model implemented in this work is able to extract actions adequately for many well-constructed sentences from experimental procedures. Although we compare it with the machine-learning model, it is not to be understood as a baseline to outperform, but rather as a stepping stone that helps us train the machine-learning model more rapidly and with less data.

The evaluation of the machine-learning model on the annotation test set results in a perfect match of the action sequence for 60.8% of the sentences. A detailed inspection of the incorrect predictions reveals that the errors are often minor (pertaining to only one action property out of the whole action sequence) and that in many cases the predicted action sequence would be an acceptable alternative to the ground truth.

Improving the automated extraction of action sequences is an ongoing effort, involving refinement of the rules to generate data for pretraining the deep-learning model and annotation of more samples for refining it. A future strategy for the selection of the sentences to annotate will be to choose the ones that the deep-learning model is least confident about.

Although we focused on experimental procedures for organic chemistry extracted from patents, the approach presented in this work is more general. It can be adapted to any extraction of operations from text, possibly requiring new training data or the definition of new action types to cover other domains adequately. Provided adequate changes to the training data and action definitions, the approach can for instance be extended to other sources, such as experimental sections from scientific publications, as well as other fields, such as solid-state synthesis.

## Methods

**Experimental procedure data**. As a source of experimental procedures, we selected the Pistachio dataset, version 3.0[30]. This dataset contains information related to more than 8.3 M chemical reactions, 6.2 M of which are associated with an experimental procedure.

For each reaction, the Pistachio dataset also contains other information such as patent details and reaction classes, as well as information extracted from the experimental procedures.

**Rule-based model derived from Pistachio**. For each experimental procedure, the Pistachio dataset contains a list of actions and associated information, extracted from the text with a combination of LeadMine[13] and ChemicalTagger[16]. Accordingly, the action types used in Pistachio are similar to the ones in Table 1. The information associated with the Pistachio actions is not operation-specific; the set of properties is common to all action types. It consists, most importantly, of a list of compounds and associated quantities, as well as fields for the temperature, duration, or atmosphere. To convert these actions to our format, we map, where possible, the action types, and post-process the data attached to these actions. For instance, each compound attached to a `Heat` action in Pistachio is converted to an `Add` action that is prepended to the `Stir` or `SetTemperature` action.

This approach to the generation of actions from experimental procedures is a good starting point, but limits us to the information detected by Pistachio and reported in the dataset. In particular, some actions relevant to us are not detected, such as all pH-related operations. Also, the Pistachio dataset contains no information about the relationships between compounds in a sentence.

**Custom rule-based NLP model**. We developed a custom rule-based natural language processing (NLP) algorithm for the extraction of operations with associated chemical compounds, quantities, and reaction conditions from experimental procedures.

In a first step, the algorithm processes a text independently of the actions defined in Table 1. It detects operations by searching for verbs corresponding to synthesis operations, defined in a custom list. By analyzing the context of these verbs, the algorithm determines the associated compounds and quantities, as well as additional operation conditions. It also identifies the role of the compounds in the sentence (subject, direct object, etc.), and the relationships between compounds.

In a second step, the operations and associated information are post-processed to map them to the action types of Table 1. This post-processing is similar to the one of the Pistachio-derived actions detailed above. For this task, information about the relationships between components and their role in the sentence are very useful. For instance, they indicate in what order compounds must be added, independently of what comes first in the sentence (for instance, "To X is added Y" or "Y is added to X" are equivalent). Also, it allows us to group compounds and convert them to `MakeSolution` actions when they belong together in the text (as in "A solution of X in Z is added to a solution of Y in Z.").

This approach to the extraction of actions from text is more flexible for our purposes than deriving the actions from Pistachio, since it can easily be modified or extended. In addition, it allows us to ingest experimental procedures from other sources than the Pistachio dataset.

**Combined actions from rule-based models**. Starting from a single experimental procedure, both rule-based approaches described above will generate two sequences of actions that may be different. An analysis of the generated actions rapidly uncovers their respective strengths and shortcomings. On the one hand, in our experience, the Pistachio-generated actions are better at extracting `Yield` actions, or at detecting under what atmosphere reactions are conducted. Our custom NLP approach, on the other hand, can cover a broader vocabulary of operations, and supports `MakeSolution` actions.

Combining both sources has the potential to generate actions that are better than each of the approaches taken separately. Formulating an algorithm to accomplish this in a clever way, however, is not straightforward. In this work, the combined dataset appends `Yield` actions from the Pistachio-based extraction to the actions generated by our custom NLP algorithm.

**Annotations**. To improve on the quality of training data based on the rule-based models, we generated higher-quality action sequences by manually annotating sentences from experimental procedures.

We developed an annotation framework based on the `doccano` annotation tool[34]. Annotators can open the framework in a web browser and navigate through sentences from experimental procedures. The page shows the sentence to annotate and a readable representation of the actions associated with it. An annotator can add new actions, reorder them, or edit them by opening a separate view. Figure 4 illustrates what a user of the annotation framework sees.

The annotation framework is pre-loaded with samples that are pre-annotated by combining action sequences from both rule-based models. The samples to annotate are sentences (from randomly picked experimental procedures) for which the rule-based extraction of actions encounters difficulties, such as sentences containing highly context-dependent verbs, sentences containing "followed by", which the rule-based models usually struggle with, or sentences that result in multiple actions referring to the same compound.

To ensure consistency among the annotators, a detailed annotation guideline was provided. It can be found in the Supplementary Data 3. Furthermore, a single annotator reviewed all the annotations.

**Data augmentation**. Data augmentation on the set of annotated samples increases the number of data points available for refinement in order to minimize overfitting. We augment the data by substituting compound names and quantities, as well as durations and temperatures, with a probability of 50%. The substitutes are selected at random from lists that we compiled from a subset of the Pistachio dataset. An example of data augmentation is shown in Table 6.

**Machine-learning model**. We formulate the extraction of action sequences from experimental procedures as a sequence-to-sequence translation, in which experimental procedures are translated to the textual representation of the actions defined in Table 1.

Restricting the output to a textual form is no limitation, since the textual representation of actions can easily be converted back to the action type and associated properties without loss. Furthermore, doing so allows for an easier and more flexible setup than designing a custom architecture for sequential prediction of actions and corresponding properties; this also means that established model architectures for sequence-to-sequence translation can be applied with few modifications.

Experimental procedures usually contain very few cross-sentence dependencies. We therefore translate experimental procedures sentence by sentence. This simplifies the learning task and limits the requirements on the model architecture. In the few cases where knowledge of the neighboring sentences would be relevant, the missing information can normally be determined from the context as a post-processing step when combining the sentences. As an example, from the sentence

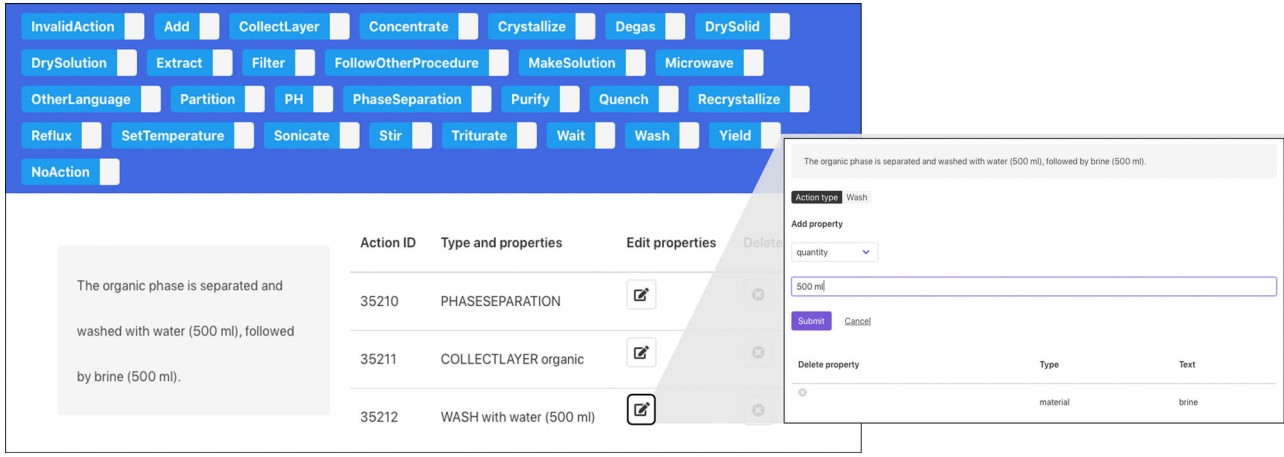

**Fig. 4 Screenshots for adding and editing actions with the annotation framework.** The sentence to annotate is displayed on the left-hand side, with the corresponding pre-annotations on the right-hand side. A `Wash` action is missing and can be added by clicking on the corresponding button at the top. Also, when clicking on the appropriate button, a new page open to edit the selected action.

---

**Table 6 Illustration of the data augmentation approach.**

**Diisopropylazodicarboxylate (0.05 ml, 0.302 mmol) was added to the reaction mixture followed by stirring for 3 h at room temperature.**

(1) *2-(2-hydroxyphenyl)-ethanol* (*0.119 mole*, 0.302 mmol) was added to the reaction mixture followed by stirring for 3 h at *7 ℃*.
(2) Diisopropylazodicarboxylate (0.05 ml, *11.65 g*) was added to the reaction mixture followed by stirring for *2 additional minutes* at 100–105 ℃.
(3) *isobutylene gas* (*24.94 mmol*, 0.302 mmol) was added to the reaction mixture followed by stirring for 3 h at room temperature.
(4) *n-methyl-4-nitroaniline* (0.05 ml, *4.57 mmol*) was added to the reaction mixture followed by stirring for *9 h* at −5 ℃.

A reference sentence (at the top) is augmented to produce four additional sentences. The substituted elements are written in italic. For data augmentation of the annotation dataset, the actions associated with the reference sentence are also subjected to substitution.

---

"The solution mixture is filtered and concentrated.", it is clear that the filtrate is kept rather than the precipitate. For "The solution mixture is filtered. It is then concentrated.", this fact can be inferred by noticing that the `Filter` action is followed by a `Concentrate` action, which indicates that the phase to keep after filtration must be the filtrate.

The deep learning model for the conversion of experimental procedures to action sequences relies on the transformer architecture[35], which is considered to be state-of-the-art in neural machine translation. To be more specific, our model uses a transformer encoder–decoder architecture with eight attention heads. The model is trained by minimizing the categorical cross-entropy loss for the output (sub)words. The model is implemented with the `OpenNMT-py` library[36,37]. The library indicates that the transformer model is very sensitive to hyperparameters and suggests a set of default parameters, which we adopted with a few changes. First, we reduced the model size by decreasing the number of layers from 6 to 4, the size of the hidden states from 512 to 256, and the size of the word vectors from 512 to 256. Second, we changed the values of the parameters `max_generator_batches` to 32, `accum_count` to 4 and `label_smoothing` to 0. Third, we chose the source and target vocabularies to be identical, and accordingly our model shares their embeddings. These changes were motivated by experiments on the pretraining task. In particular, the reduction in model capacity led to a model that is easier and faster to train without considerable impact on the model performance observed with the validation set. The `OpenNMT-py` configuration file for pretraining, containing all the hyperparameters, is available as the Supplementary Data 4.

The translation model is pretrained on the action sequences generated by combining the NLP and Pistachio approaches. We apply the algorithm to a random subset of 1.0M experimental procedures, which produces 4.66M pairs of sentences and action sequences. To avoid biases due to incorrectly assigned `InvalidAction` and `NoAction`, all the `InvalidActions` are removed, as well as the `NoActions` that are longer than 30 characters and do not contain any keyword related to compound analysis. This provides more than 4.05M pairs of sentences and corresponding action sequences. After removal of duplicate sentences, 2.76M samples are remaining, which are split into training, validation, and test sets of size 2.16M, 0.27M, and 0.27M, respectively.

A vocabulary of size 16,000 is created from the training set with the `SentencePiece` library[38,39]. The source and target strings are then tokenized using the corresponding `SentencePiece` tokenizer. The model is then pretrained for 500,000 steps.

A total of 1764 annotated samples are split into training, validation and test sets of size 1060, 352, and 352, respectively. Based on this data, training is continued for the final model of the pretraining step. Three experiments are run. In the first experiment, the training set containing 984 samples is used as such ("no augmentation"). In the

second experiment, the dataset is augmented as described above to produce 20,000 samples ("augmented"). In the third experiment, the duplicates contained in the augmented dataset are removed, which results in 14,168 samples ("augmented unique"). The validation and test sets are not augmented.

Each of the three refinement experiment is repeated three times with different random number generator seeds. All the models are refined for 30,000 steps, with checkpoints saved every 1000 steps. For analysis, we then select the model checkpoint leading to the highest accuracy. Some of the models selected in this fashion are combined into ensemble models. Additionally, three models are trained on the annotated dataset only (no pretraining).

While the different splits (training, validation, test) of the pretraining and annotation datasets contain strictly different sentences, we note that the language of experimental procedures is limited and many sentences will therefore not differ very much. This overlap, however, is difficult to measure and to avoid.

## Data availability

The data on which the models for the extraction of action sequences were trained are available from NextMove Software in the Pistachio dataset[30]. The rule-based and hand-annotated action sequences are available from the authors upon request.

## Code availability

A Python library with the action definition and handling as well as associated scripts for training the transformer model can be found on GitHub at https://github.com/rxn4chemistry/paragraph2actions. The trained models can be freely used online at https://rxn.res.ibm.com or with the Python wrapper at https://github.com/rxn4chemistry/rxn4chemistry to extract action sequences from experimental procedures.

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

## Acknowledgements

We thank the anonymous reviewers for their careful reading of our manuscript and their many insightful comments and suggestions.

## Author contributions

The project was conceived and planned by T.L. and A.C.V. and supervised by T.L. F.Z. designed the custom rule-based NLP model. A.C.V. implemented and trained the other models. J.G. set up the annotation framework. All the authors were involved in discussions about the project and annotated the dataset. A.C.V. reviewed all the annotations and wrote the manuscript with input from all authors.

## Competing interests

The authors declare no competing interests.
