## [Peer Review File · Nature Communications]

Reviewers' comments:

Reviewer #1 (Remarks to the Author):

This work by Laino and coworkers looks at the problem of extracting chemical synthesis actions from procedures written in English prose. The authors use a Transformer architecture to learn to reproduce a rule-based parsing strategy and then refine the model on manually-annotated samples. The manuscript describes some considerations for the task of action sequence extraction. However, I am concerned that it is not a significant contribution to the field given the lack of open source code, lack of transparency regarding the rule-based models, lack of open source annotated data, and lack of a new methodological advance. It's a nice application of the Transformer and a worthwhile task, but I wouldn't say that it advances the state of the art given the near-complete lack of reproducibility.

1. The introduction needs to be much more precise to convey what has been the goal of previous approaches and how their information extraction procedures operated. Right now, the introduction provides a cursory description of related work, but it's not clear from the text what previous studies have accomplished. The introduction would be much improved if additional time was spent describing previous contributions [3] and work that has been done in inorganic chemistry [9-11]; specifically, describing the methods that these approaches used rather than just the application/context for them.
2. "more simple" inorganic synthesis probably warrants clarification – are there fewer actions to define? Is it easier to recognize named entities? Are paragraphs written with greater regularity?
3. NER is described as "allow[ing] the detection of compounds, actions, or experimental conditions in texts from the chemistry literature". To make sure that readers unfamiliar with information extraction understand what these tasks entail, it would be better to be precise about how these tasks are defined.
4. If Reaxys is mentioned, the authors should probably describe what their process of information extraction is and the type of data they capture.
5. "This approach is exposed to linguistic challenges that makes a correct interpretation difficult." What does this sentence mean?
6. "Here, we present a robust model to convert experimental procedures into structured action sequences that do not require human verification." – this gives little indication of what the approach is, how it differs from previous methods. Moreover, it's rather bold to claim that the output of any modeling effort does not require human verification, especially when the accuracy is only ~65%.
7. Did the authors take inspiration from any previous information extraction efforts when constructing their list of action types? It seems like the answer is a strong yes, from Pistachio.
8. Is there a reason that "until ____" phrases were not included as an indication of timing in developing the rule-based system if it is a known limitation?
9. To clarify the full-sentence accuracy, are the authors evaluating whether the sequence of actions and related values/entities are correctly selected? Or just that the list of actions (Add, Add, Mix, Heat, etc.) are correct?
10. It would be nice to see some examples of the procedures, what is extracted by the rule-based system, and what is extracted by the ensemble model to illustrate the value (beyond the quantitative accuracy) in the main text. I know there is one example currently, and I appreciate that

it will be difficult to find concise examples, but most readers probably won't open the supplemental txt files.

11. The claim that the authors approach can be extended to other sources should be qualified or elaborated upon. Nowhere are synthesis procedures described more clearly than in these organic synthesis patents.

12. I appreciate referring to the Methods section for details, but this study's significant reliance on Pistachio warrants some mention in the main text.

13. The ~35% incorrect predictions of the ensemble model are partially justified by the authors, but what about the ~75% incorrect predictions of the rule-based model? Are these errors for similar reasons? Some of the errors of the RB system in the SI are a little confusing to me. As the major contribution of this work is a tool with higher accuracy, it is probably worth explaining in greater detail how the RB system operates and why it makes the mistakes that it does to ensure that it's not an artificially poor baseline. Some examples:

a. "The reaction mixture was cooled to 0° C., and toluene (100 ml) and water (150 ml) were successively added dropwise." – why would a RB system not start with the SETTEMPERATURE if it comes first in the sentence?

b. "The mixture was left to stir at rt. for 1 h." – why would a RB system not recognize the word "stir" to indicate the "STIR" action?

c. "This mixture was left to stand, well sealed, for 24 hours at 20°C." – why would a RB system not interpret "left ... for 24 hours at 20 C" as a WAIT?

14. Were multiple annotated samples taken from the same document? If one patent has an embodiments section that describes many similar reactions, it will use virtually identical language for all of those paragraphs. If yes, and if the 1764 annotated samples were randomly split, then there will be a great deal of overlap between the training and testing action sequences. The evaluation should be done so that test samples come from unseen documents.

15. No justification is provided or changing hyperparameters in the OpenNMT-py library. Was a formal hyperparameter optimization performed?

16. The lack of code availability (for the rule-based method and for the Transformer-based method) is very disappointing. I know this isn't required for Nature family journals, but it's a shame. Trained model ability through a web application is not the same.

17. Are the authors planning to release the 1764 annotated examples?

Reviewer #2 (Remarks to the Author):

The manuscript titled "Automated extraction of chemical synthesis actions from experimental procedures" describes a machine learning workflow for translating synthetic organic experimental procedures into a series of actions. The authors define a set of actions that should broadly encompass most batch chemistry reactions. Furthermore, they build a machine learning model based on the OpenNMT transformer framework to translate sentences to action sequences. The work is a start in automating the translation of procedures into a machine-executable set of sequential actions that could be used by robotic platforms.

The manuscript is recommended to be published after some minor revisions. With the minor

revisions, the manuscript should appeal to a broad audience and is an advance toward linking historical procedure data to automated synthesis execution.

Introduction:

- The authors should make it more explicit that this procedure deals with batch chemistry since there are other types such as flow and mechanochemistry. The current actions would not cover these other types.
- The authors cite reviews ref 6 and 14 but should pull the relevant citations from the review to properly acknowledge the work that is being discussed in the intro.
- It is not entirely clear who the targeted audience is for the manuscript. If a broad audience is targeted then: Chemists will desire more full procedure action sequences (possibly in a SI) similar to table 2 that are translated by the model. Informaticians will want more statistics about the data (discussed below) and the results (was this trained once or is this an average of multiple trainings with different random splits). The machine learning audience will desire a better rationalization of why the transformer model is the correct choice and at least a description of the model (what is the objective function, how many attention heads, etc..)

Results:

- Table 1: CollectLayer – Not always aqueous and organic partitioning. Sometimes can be organic-organic such as methanol hexanes. Specify if this would be captured by this action.
- Table 1: Crystallize – Co-Solvents or just single solvent?
- Table 1: Filter – does this include filtration through agents such as Celite?
- Table 1: Partition and extract seem the same. Rationalize why both are needed.
- Table 1: Reflux – if refluxing a solvent, one is keeping the temperature at the BP of said solvent by condensing back into flask. Is the extra field necessary when it could be included in SetTemp by looking up BP's?
- A full description on how the BLEU score was modified is necessary for the reader to gain a full understanding of the results.

Methods:

- A better description and analysis of the data would be helpful for the reader to gain an appreciation of the translation task. Some statistics on the data distribution, lengths of sentences, average number of actions in a sentence, the distribution of actions (ie which are the most common) etc.. would be helpful.
- The Pistachio dataset contains many duplicate entries, where the patent is filed in multiple locations such as US and Europe. How many of the datapoints are duplicates? If those were filtered out, then please add to the description. From the reviewers experience, less than half of the 8.3 million are unique.
- A brief description of how rule based model 1 works (if possible to get from nextmove) would be helpful for the reader to better understand the differences between the two rule based models.
- Similar to the analysis of the Pistachio dataset stated above, an analysis of the annotated dataset would be helpful. How do the two datasets differ in their distribution of actions. Does the annotated dataset contain longer sentences, a larger distribution of actions etc.. Presumably, they are similar but since the datasets are not free for the reader to access, this should be included. Additionally, the small size (~350 data points) of the annotated test set could not be very diverse which would be a cause of concern.

Machine Learning Model:

- There are many neural translation models and a rationalization of why the authors chose the transformer would be good. OpenNMT also contains other seq-2-seq models that are ready to use out of the box so a short (non hyperparameter optimized) comparison would make the choice seem less more rationalized.
- A full list of all the hyperparameters used (in the main text or SI) is necessary. The model defaults may change over time in the OpenNMT github and this would improve the ability for others to reproduce the work.
- Is the vocabulary size actually 16k or is that the maximum allowed? If it is the max allowed then do the sentences actually contain 16k where then the list is truncated, or less?

SI:

- In Note 2 acknowledge that these actions cover most batch chemistry procedures but in the current form (which can easily be extended) does not capture some emerging chemistries such as electrochemistry (voltage, wattage, electrode type, etc..) and light mediated transformations (wavelength). The reviewer believes that these will be desirable in robotic execution for difficult to access structures.

Other:

- There is a full reference and DOI for OpenNMT technical report (<https://www.aclweb.org/anthology/P17-4012/>) which should be used in place or in addition to ref 26
- When discussing datasets such as Reaxys, include Scifinder (CAS) since it is also a human curated dataset of comparable size and diversity.
- How does model handle “general procedures” where no specific reagent(s) are given. This is often seen in procedures where the same reaction is performed on multiple similar substrate. It seems the actions would still be translated but a short sentence acknowledging that this would not be useful for robotic execution without specific chemicals is warranted. eg:
Specific Procedure: To a suspension of methyl 3-7-amino-2-[(2,4-dichlorophenyl)(hydroxy)methyl]-1H-benzimidazol-1-ylpropanoate (6.00 g, 14.7 mmol) and acetic acid (7.4 mL) in methanol (147 mL) was added acetaldehyde (4.95 mL, 88.2 mmol) at 0° C...
General Procedure: To a suspension of the benzimidazole (6.00 g, 14.7 mmol) and acetic acid (7.4 mL) in methanol (147 mL) was added the aldehyde (4.95 mL, 88.2 mmol) at 0° C...
- The sentence “In the second experiment, the dataset is augmented as described above to produce 20000 samples (“augmented”)” the details of how the data is augmented are not clear or where above it is described.
- It is recommended to open source the code so that it can be reproduced by researchers that have access to Pistachio. From the text, it does not appear that much of the code is proprietary (OpenNMT and sentencepiece are already open) so putting together the pipeline and publishing it on github should not be an obstacle.

Reviewer #3 (Remarks to the Author):

In this paper the authors propose an approach for automatically extracting experimental procedures for organic chemical synthesis from English text, (specifically, the experimental procedures from patents). Their approach consists of annotating raw text sentences with corresponding sequences of synthesis actions and their parameters, then training a supervised neural network model, originally developed to perform the task of machine translation (automatically converting human-language sentences from one language to another), to output the correct sequence of actions.

This paper describes a solid contribution in the area of scientific information extraction / chemical informatics. Their main technical contribution is a manually annotated dataset of experimental procedures (1764 sentences), including an ontology for chemical synthesis steps and their parameters. They show that an ensemble of off-the-shelf machine translation models can be trained on this data to achieve whole-sentence accuracy of 64.5% (i.e. 64.5% of sentence-level action sequences in the test set of 352 such sequences were exactly reproduced by the model). They also suggest that pre-training the model with weak supervision from rule-based extractions is potentially useful in this setting, though this is not demonstrated in the experimental results, since no results are reported for a model lacking this pretraining, or comparing to a simpler baseline such as pretraining on the raw text with a self-supervised language modeling objective (e.g. BERT (Devlin et al. 2018)).

In addition to more/better baseline models, I would like to see a more detailed quantitative analysis of experimental results. Most reported analysis is qualitative, but it would be interesting to see model performance broken down by e.g. action label. This would be more feasible under a different evaluation paradigm (proposed in more detail below). I would also like to see a subset of examples annotated by different annotators in (at least) duplicate in order to report inter-annotator accuracy (e.g. Fleiss' kappa), as is standard, to measure the quality/consistency of the annotation. There are many cases where the mapping from text to actions is potentially ambiguous.

Some of the most important limitations of this work are the annotation schema's inability to capture nonlinear experimentation, e.g. when a product is divided and subsequently used in two different experimental pathways, or connecting to an earlier described procedure. These limitations are described in the paper but I would like to see more quantification justifying these decisions, such as the statement that "experimental procedures usually contain very few cross-sentence dependencies." As it stands I'm concerned that the proposed data represent only a substantially simplified subset of the actual problem, and thus have limited potential for real-world impact.

Another concern I have is regarding evaluation. Both BLEU score and whole-sentence accuracy compare the exact order of the series of experiment actions and parameters converted to text. This evaluation potentially unfairly favors the authors' proposed technique, which is adapted from machine translation and is thus trained to produce the sequence of actions and their parameters converted to natural language text. However, a new system implementing a more structured

approach might produce the same annotations as an ordered list of actions and (importantly, potentially unordered) parameters for each action. In order to be directly comparable to the proposed approach, this new system's output would then have to be converted to text as well, potentially introducing errors in this superficial text conversion process that do not correspond to errors in the actual action sequence. For example, naive conversion might result in differently ordered parameters for an action, which unfairly favors models trained to explicitly produce action parameters in a certain order, despite this being an irrelevant artifact of the conversion to text. Indeed, a model that is trained to be invariant to these orderings might be more robust to variations in the text, but would potentially be penalized under BLEU or whole-sentence accuracy as proposed. An alternative approach that would resolve my concerns would align predicted actions with the gold-standard output to predict action accuracy (using e.g. Levenshtein distance), then for each correctly-aligned action compare order-invariant extracted parameters. BLEU is a poor approximation (by n-gram overlap) of this more exact evaluation that was designed for the much more ambiguous application of machine translation (where, unlike here, there may be many different correct translations), where it has also been shown to have issues such as poor correspondence to human acceptability judgements.

More detailed notes for the authors:

- Would like to know earlier in the text (abstract or introduction) what the neural network model is going to look like (encoder-decoder/ seq2seq). "transformer" is often used to just refer to the multi-head self-attention (versus the original encoder-decoder application of it).
- Would be interesting to see discussion of information required for a human to successfully conduct an experiment versus a machine — certainly a human has a great deal of practical knowledge on experimentation that will not be explicitly stated in the text, resulting in extractions that are actionable for a human but far from actionable by a machine without an additional great deal of knowledge being encoded/curated from somewhere. How do you address this missing (implicit) information in your annotation scheme or model? For example, the text may simply say "dry", implying the DryInVacuum operation since the product is a solid. "FollowOtherProcedure" does this to some extent. There is discussion of implicit actions, which is handled by a sentence with e.g. only one verb being mapped to many actions, or heat may go to "set temp" in some cases or "stir" in another. How does the model perform on these types of examples? Do you have annotation guidelines for handling these consistently? What is your inter-annotator agreement?
- How does XDL compare to the output of your model? Is substantial post-processing, addition of implicit information required to map from your annotations to XDL?
- How do you deal with branched synthesis procedures in your data? Are they simply ignored and not annotated?
- Breakdown by action types? e.g. MakeSolution seems hard. How often does InvalidAction happen, and is it included in your performance metrics? Is NoAction included in performance evaluation?
- You report that a single annotator reviewed all the annotations, but it is standard to have multiple annotators annotate at least a subset of the data in order to report inter-annotator accuracy. As it stands we have no indication of annotator agreement, and thus consistency of annotations, for the dataset.
- I don't know if I agree that "experimental procedures usually contain very few cross-sentence dependencies." Could you quantify this? Perhaps in your limited framework where syntheses are linear, references to other parts of the text, and actions that depend on state are not really

captured, cross-sentence dependencies that fit into your framework are rare. But I think it's quite common to see e.g. a product from a previous sentence referenced in a subsequent sentence, and making these connections would be very important in nonlinear experiments.

- Why did you reduce the model size and remove label smoothing?
- Did you try just ensembling models with different random seeds, versus the augmented vs not augmented models? Could work just as well / better.
- How did you decide when to stop pretraining (at 350k steps)?
- You may also want to cite: <https://www.aclweb.org/anthology/N18-2016/>

Typos/grammar/style:

- Introduction:
 - while scientists developed -> while scientists have developed
 - "Utilizing inherent rules": not sure inherent is the word you want here — the rules are a heuristic attempting to get at some inherent properties specific to the data
 - Important human effort -> substantial human effort
 - Not sure what "deep search" is
 - Reviving use of robots, as in they were once used, then less so, and now mores anew?
- Results
 - Would be great to see an example structured synthesis resulting from the example text at the beginning of results section. Ah! Table 2 should come earlier, much closer to the example text.
 - You need to make it more clear what "augmentation" and "refined" mean in this table. I understand that refined is fine-tuning with manually annotated data, though that wasn't completely obvious to me. Augmentation is not described until the methods section.
- Methods
 - Figures 1 and 2: I don't think you need an image (and especially not two) depicting the annotation framework; space could be better spent providing more quantitative analysis. This would be appropriate if the main contribution of your work were the annotation tool itself but I don't think that's the case.

Reviewer #1 (Remarks to the Author):

This work by Laino and coworkers looks at the problem of extracting chemical synthesis actions from procedures written in English prose. The authors use a Transformer architecture to learn to reproduce a rule-based parsing strategy and then refine the model on manually-annotated samples. The manuscript describes some considerations for the task of action sequence extraction. However, I am concerned that it is not a significant contribution to the field given the lack of open source code, lack of transparency regarding the rule-based models, lack of open source annotated data, and lack of a new methodological advance. It's a nice application of the Transformer and a worthwhile task, but I wouldn't say that it advances the state of the art given the near-complete lack of reproducibility.

We take note of the referee's concerns. We address the points raised by the referee in the comments below and hope to be able to convince him, with the updated version of this manuscript, that this is an adequate contribution for Nature Communications.

Our work does advance the state of the art (see, for instance, the comparison in <https://twitter.com/ForRxn/status/1213115635489746944?s=20>, or the value of 56% of extracted event arguments in <http://arxiv.org/abs/1711.06872>), and we think that the application of a translation model to an extraction task with a structured output, together with the two-step training procedure, is innovative and can be of interest to a broad audience.

1. The introduction needs to be much more precise to convey what has been the goal of previous approaches and how their information extraction procedures operated. Right now, the introduction provides a cursory description of related work, but it's not clear from the text what previous studies have accomplished. The introduction would be much improved if additional time was spent describing previous contributions [3] and work that has been done in inorganic chemistry [9-11]; specifically, describing the methods that these approaches used rather than just the application/context for them.

We rewrote the part of the introduction mentioning related work. We provide more detail about the relevant approaches. We do think that this results in a considerable improvement, even though it is not possible, for space reasons, to explain the methods for all the previous contributions in detail.

2. "more simple" inorganic synthesis probably warrants clarification – are there fewer actions to define? Is it easier to recognize named entities? Are paragraphs written with greater regularity?

Indeed, we did not justify this statement. We were referring to the usually shorter synthesis routes and the fact that for many syntheses, subsets of operations are sufficient ([dx.doi.org/10.1038/s41524-017-0055-6](https://doi.org/10.1038/s41524-017-0055-6)). Since a detailed comparison between inorganic and organic syntheses is not relevant for the present work, we removed this statement from the manuscript.

3. NER is described as "allow[ing] the detection of compounds, actions, or experimental conditions in texts from the chemistry literature". To make sure that readers unfamiliar with information extraction understand what these tasks entail, it would be better to be precise about how these tasks are defined.

We adapted our description of NER and of the relevant work in chemistry.

4. If Reaxys is mentioned, the authors should probably describe what their process of information extraction is and the type of data they capture.

With Reaxys being a commercial product, it is not easy to find information about their pipeline for extracting information. In the manuscript, we now mention the data curation by experts, and also refer to SciFinder (see comment by Referee #2).

5. “This approach is exposed to linguistic challenges that makes a correct interpretation difficult.” What does this sentence mean?

Linguistic noise in the experimental procedures can lead to misinterpretation. The presence of incorrectly formulated sentences creates even more challenges for any natural language tool such as the tool to create XDL schemas. The authors themselves state, in the supporting information: “This process has obvious room for misinterpretation and error depending on how the experimental procedure is written, therefore human verification of the translation is recommended.”

We updated the text in the manuscript to clarify the statement.

6. “Here, we present a robust model to convert experimental procedures into structured action sequences that do not require human verification.” – this gives little indication of what the approach is, how it differs from previous methods. Moreover, it’s rather bold to claim that the output of any modeling effort does not require human verification, especially when the accuracy is only ~65%.

We agree that these statements can seem contradictory. As has now been made clearer in the manuscript, the performance of the model is better than the value of 65% suggests. We refer to the new Table 5 and to the Supplementary Data 1 to illustrate this.

We modified this sentence to state the avoidance of human verification as a goal rather than a claim.

7. Did the authors take inspiration from any previous information extraction efforts when constructing their list of action types? It seems like the answer is a strong yes, from Pistachio.

We agree that the actions defined in Pistachio deserve some credit in this regard (they were originally defined in “ChemicalTagger: A tool for semantic text-mining in chemistry”, DOI: 10.1186/1758-2946-3-17).

We updated the manuscript to mention that the above paper also defined actions based on patents, and that other papers also selected action types in the context of solid-state synthesis.

8. Is there a reason that “until ___” phrases were not include as an indication of timing in developing the rule-base system if it is a known limitation?

The “until ...” phrases do not fit in the action schema, since they do not specify what is done, but rather until when something else is done. Technically, including the “until ...” phrases by adding a corresponding property to some action types would be feasible, and we are confident that the transformer model would be able to extract the corresponding part of the text. However, storing this information as a raw string would miss the purpose of the current work, as it would make a non-trivial post-analysis of that string necessary. One exception that is currently supported (and also common) is when, in the experimental procedure, a

compound is added until a given pH is reached. In this case, the pH value is directly extracted as a property of the PH action.

9. To clarify the full-sentence accuracy, are the authors evaluating whether the sequence of actions and related values/entities are correctly selected? Or just that the list of actions (Add, Add, Mix, Heat, etc.) are correct?

The full-sentence accuracy measures perfect matches between the predicted action sequence and the ground truth. I.e., we check that all the actions, compounds, quantities and any related values/entities are correctly extracted. This also explains why the accuracy is “only” 65%: often, the correct action type is predicted but with incorrect properties. To illustrate this better, in the updated manuscript we avoid the term "full-sentence accuracy". Instead, we report the fraction of perfect (100%) matches, as well as the fractions of 90% matches and 75% matches.

We also refer to the new Table 5, which distinguishes between correctly predicted action types and correctly predicted actions including associated values (see detailed comment further below).

10. It would be nice to see some examples of the procedures, what is extracted by the rule-based system, and what is extracted by the ensemble model to illustrate the value (beyond the quantitative accuracy) in the main text. I know there is one example currently, and I appreciate that it will be difficult to find concise examples, but most readers probably won't open the supplemental txt files.

We added Table 4 which contains 5 examples of sentences with the corresponding action sequences as annotated and as predicted by the machine-learning model. This table not only provides an illustration of how and when the actions are used, but also gives some insight into a few differences between the ground truth and the predicted actions.

11. The claim that the authors approach can be extended to other sources should be qualified or elaborated upon. Nowhere are synthesis procedures described more clearly than in these organic synthesis patents.

Our full workflow (rule-based model for pre-annotation, human annotation step and the final training of the model) is general and can tackle generic documents. We updated the manuscript to clarify this.

12. I appreciate referring to the Methods section for details, but this study's significant reliance on Pistachio warrants some mention in the main text.

We agree and now mention it in the main text when introducing the models. The algorithms used to extract data in Pistachio (ChemicalTagger, LeadMine) are now also more explicitly cited in our manuscript.

13. The ~35% incorrect predictions of the ensemble model are partially justified by the authors, but what about the ~75% incorrect predictions of the rule-based model? Are these errors for similar reasons? Some of the errors of the RB system in the SI are a little confusing to me. As the major contribution of this work is a tool with higher accuracy, it is probably worth explaining in greater detail how the RB system operates and why it makes the mistakes that it does to ensure that it's not an artificially poor baseline. Some examples:

- a. “The reaction mixture was cooled to 0° C., and toluene (100 ml) and water (150 ml) were successively added dropwise.” – why would a RB system not start with the SETTEMPERATURE if it comes first in the sentence?
- b. “The mixture was left to stir at rt. for 1 h.” – why would a RB system not recognize the word “stir” to indicate the “STIR” action?
- c. “This mixture was left to stand, well sealed, for 24 hours at 20°C.” – why would a RB system not interpret “left ... for 24 hours at 20 C” as a WAIT?

The main task for the rule-based component is to provide data for pre-training the ML model. As such, we did not intend to present it as a baseline that we want to outperform with a ML model. Rather, we wanted to show that our refinement approach works and produces a model that is better than what we could achieve using rules.

We also agree that some of the predictions by the rule-based model are clearly incorrect. Fixing these errors is possible in theory by adding more rules; in order to achieve an accuracy comparable with the ML model, however, this would incur a considerable amount of time-consuming manual work. We chose a trade-off by having a rule-based model that is qualitatively good enough and, more importantly, is able to generate adequate pretraining data.

That being said, we would like to explain the examples raised by the referee.

- a. “The reaction mixture was cooled to 0° C., and toluene (100 ml) and water (150 ml) were successively added dropwise.”
In this sentence, the rule-based model fails to recognize that “toluene” and “water” belong only to “added”, and it assigns them to the action “cooled” as well. As a rule, any compounds associated with a “cool” action will first lead to an “Add” action, which explains why the sentence does not start with SetTemperature and why the Add actions are duplicated.
- b. “The mixture was left to stir at rt. for 1 h.”
Defining rules for constructions such as “was left to stir”, “was allowed to stir”, “was permitted to stir” is more involved than “was stirred”, because the central verb is no longer the action itself. Hard-coding specific rules to cover this is possible in principle, but would also need to account for other usages of those verbs (such as: “was left to reach”, “was left stirring”, “was left to be cooled”), that feature a higher variability than when the action verbs are used directly. Instead, we decided to let the machine-learning model learn such constructions by overrepresenting them in the annotation dataset.
- c. “This mixture was left to stand, well sealed, for 24 hours at 20°C.”
This case is similar to b.

In the Discussion, we now explain that the rule-based model is mainly useful to train the ML model more effectively rather than to provide a baseline.

14. Were multiple annotated samples taken from the same document? If one patent has an embodiments section that describes many similar reactions, it will use virtually identical language for all of those paragraphs. If yes, and if the 1764 annotated samples were randomly split, then there will be a great deal of overlap between the training and testing action sequences. The evaluation should be done so that test samples come from unseen documents.

There is probably some overlap between the training and testing action sequences. This overlap is, however, very difficult to measure and avoid. Not only may the sentences be similar in a single document, as pointed out by the referee, but also in different patents from the same company or the same inventors.

We did not constrain annotated samples to be taken from different documents. With the experience we acquired in the present work, we do not believe that this overlap is large. Considering the fact that the annotated sentences come from a corpus of 1M experimental procedures (850k unique procedures), each of which consists of several sentences, we think that the effect of identical language having its origin in the same documents will be minimal for a comparatively small set of 1764 sentences.

The challenge pointed out by the referee is still very relevant: the language from different patents, while varying quite a lot, is still constrained to the topic of organic chemistry. Many sentences will therefore not differ very much except for the compound names, quantities, temperatures or durations, as well as linguistic choices (active/passive, present/past, synonyms, etc.). Therefore, we do not think that a perfectly adequate separation in training and test sets is possible. It is difficult to do more than avoid identical sentences to be present in different splits.

15. No justification is provided or changing hyperparameters in the OpenNMT-py library. Was a formal hyperparameter optimization performed?

We determined the hyperparameters based on the default hyperparameters suggested in the OpenNMT repository, and building on our own experience with transformer models from other projects (see f.i. <http://dx.doi.org/10.1021/acscentsci.9b00576>) .

Concretely, setting the label smoothing to 0 improves the performance on the pretraining task. Also, earlier experience with transformer models in related projects showed that the performance of the model remained very high when decreasing the model capacity, which we also noticed in the present work. Other than that, no additional hyperparameter tuning was done: even the OpenNMT documentation points out the challenges of doing so (“The transformer model is very sensitive to hyperparameters”).

We updated the manuscript to explain our choice of hyperparameters.

16. The lack of code availability (for the rule-based method and for the Transformer-based method) is very disappointing. I know this isn't required for Nature family journals, but it's a shame. Trained model ability through a web application is not the same.

We understand the referee's comment and we would like to stress that the trained model services (web application and Python wrapper) are already a good source for reproducing and validating the scientific claim of the paper or performing analyses of additional paragraphs. The code related to the actions and the training of the transformer-based models is now available on GitHub (<https://github.com/rxn4chemistry/paragraph2actions>). Combined with the data (rule based annotated, human annotated, etc..), which is available from the authors upon request and upon proof of a Pistachio license, a user would now be able to reproduce our results. The rule-based tools are IBM proprietary and are not available for distribution. Moreover, this is not an essential component as it is necessary only for the initial annotation.

17. Are the authors planning to release the 1764 annotated examples?

All data (rule based annotated, human annotated, etc..) are available upon request as described above.

Reviewer #2 (Remarks to the Author):

The manuscript titled “Automated extraction of chemical synthesis actions from experimental procedures” describes a machine learning workflow for translating synthetic organic experimental procedures into a series of actions. The authors define a set of actions that should broadly encompass most batch chemistry reactions. Furthermore, they build a machine learning model based on the OpenNMT transformer framework to translate sentences to action sequences. The work is a start in automating the translation of procedures into a machine-executable set of sequential actions that could be used by robotic platforms.

The manuscript is recommended to be published after some minor revisions. With the minor revisions, the manuscript should appeal to a broad audience and is an advance toward linking historical procedure data to automated synthesis execution.

Introduction:

- The authors should make it more explicit that this procedure deals with batch chemistry since there other types such as flow and mechanochemistry. The current actions would not cover these other types.

In “Synthesis Actions”, we added a sentence to clarify that our actions apply to batch chemistry.

- The authors cite reviews ref 6 and 14 but should pull the relevant citations from the review to properly acknowledge the work that is being discussed in the intro.

Ref. 6 was added by mistake instead of an article about reaction prediction by the same author. We corrected this in the manuscript.

The updated introduction (see other comments) now contains additional citations from Ref. 14, and we explicitly refer the reader to read Ref. 14 for an extensive review of methods for chemical named entity recognition.

- It is not entirely clear who the targeted audience is for the manuscript. If a broad audience is targeted then: Chemists will desire more full procedure action sequences (possibly in a SI) similar to table 2 that are translated by the model. Informaticians will want more statistics about the data (discussed below) and the results (was this trained once or is this an average of multiple trainings with different random splits). The machine learning audience will desire a better rationalization of why the transformer model is the correct choice and at least a description of the model (what is the objective function, how many attention heads, etc..)

Indeed, we do target a broad audience for this manuscript.

Chemists:

We added Table 4, which lists a few additional examples of sentences and corresponding action sequences (also suggested by Referee #1). It provides an illustration of how and when the actions are applied, in addition to showing a few differences between the model prediction and hand annotations. Additional examples can be found in the Supplementary Data 1.

Informaticians:

We added a subsection called "Data Insights" and extended our discussion of the results to provide a better illustration of the kinds of errors made by the machine-learning model. We also improved the description of the training process.

Machine learning audience:

We updated the manuscript to better explain our choice of architecture and discuss the hyperparameters in more detail.

Results:

- Table 1: CollectLayer – Not always aqueous and organic partitioning. Sometimes can be organic-organic such as methanol hexanes. Specify if this would be captured by this action.
- Table 1: Crystallize – Co-Solvents or just single solvent?
- Table 1: Filter – does this include filtration through agents such as Celite?
- Table 1: Partition and extract seem the same. Rationalize why both are needed.
- Table 1: Reflux – if refluxing a solvent, one is keeping the temperature at the BP of said solvent by condensing back into flash. Is the extra field necessary when it could be included in SetTemp by looking up BP's?

CollectLayer – An organic-organic partitioning would not be detected, as both methanol and hexanes would be considered organic. We added this to the list of limitations for the current set of actions in the Supplementary Note 5.

Crystallize – This action type is also used for co-solvents. This is for instance the case in one of the test set sentences available in the Supplementary Data 1: "[...] which was recrystallised from dichloromethane/petrol [...]" is converted to "RECRYSTALLIZE from dichloromethane/petrol". The same happens in other actions, such as additions, that involve already-existing mixtures of solvents. We updated Table 1 to clarify this.

Filter – This action only cares about the phase to keep and not about the specifics of the filters used.

Partition and extract – In our definition Partition indicates the addition of two immiscible solvents to the reaction mixture and will most likely be followed by a CollectLayer action. Extract, on the other hand, involves the addition of only one solvent, and implicitly considers the selection of the corresponding phase from the biphasic mixture for further processing. We updated Table 1 to clarify this.

Reflux – We do not include such a property for the Reflux action: in experimental procedures, the solvent and temperature are rarely specified in the context of reflux operations. If required for a specific application, the solvent/temperature can be determined at a later stage from the full extracted action sequence.

- A full description on how the BLEU score was modified is necessary for the reader to gain a full understanding of the results.

We added the Supplementary Note 3 to explain and illustrate the modification and we refer to it in the main text.

Methods:

- A better description and analysis of the data would be helpful for the reader to gain an appreciation of the translation task. Some statistics on the data distribution, lengths of sentences, average number of actions in a sentence, the distribution of actions (ie which are the most common) etc.. would be helpful.

We added a subsection called "Data Insights" in Results. It provides illustrations of the distribution of sentence lengths, numbers of actions per sentences, as well as action type frequencies.

- The Pistachio dataset contains many duplicate entries, where the patent is filed in multiple locations such as US and Europe. How many of the datapoints are duplicates? If those were filtered out, then please add to the description. From the reviewers experience, less than half of the 8.3 million are unique.

Indeed, in our initial submission, we did not filter out duplicates for pre-training and it turns out that slightly less than 15% of the experimental recipes were duplicates. When looking at the sentences of the generated datasets, slightly more than 30% of them were duplicates. The annotation dataset contained no duplicates by construction. For the paper revision and updated model, the duplicates were removed in the pretraining dataset. We updated the manuscript to reflect these changes.

- A brief description of how rule based model 1 works (if possible to get from nextmove) would be helpful for the reader to better understand the differences between the two rule based models.

We updated the description of this model with the available information and added the corresponding references.

- Similar to the analysis of the Pistachio dataset stated above, an analysis of the annotated dataset would be helpful. How do the two datasets differ in their distribution of actions. Does the annotated dataset contain longer sentences, a larger distribution of actions etc.. Presumably, they are similar but since the datasets are not free for the reader to access, this should be included. Additionally, the small size (~350 data points) of the annotated test set could not be very diverse which would be a cause of concern.

In the new section "Data Insights", we provide a few plots that illustrate how the action types differ between the annotation set and Pistachio, as well as between the full annotation set and its test split. There, we explain that the differences between the annotation set and Pistachio reveal the bias in the selection of samples to annotate, and that the annotation test set has a distribution close to the full annotation set despite its small size.

Machine Learning Model:

- There are many neural translation models and a rationalization of why the authors chose the transformer would be good. OpenNMT also contains other seq-2-seq models that are ready to use out of the box so a short (non hyperparameter optimized) comparison would make the choice seem less more rationalized.

We selected the transformer architecture because it is commonly recognized as state-of-the-art in neural machine translation (see, f.i., <https://www.aclweb.org/anthology/C18-1054.pdf>, <https://arxiv.org/pdf/1909.03149.pdf>, <https://ai.googleblog.com/2019/07/robust-neural-machine-translation.html>).

A comparison with the default seq2seq algorithm of the OpenNMT library on the pretraining task shows a clear superiority of the transformer-based architecture.

We updated the manuscript to better explain our choice of architecture.

- A full list of all the hyperparameters used (in the main text or SI) is necessary. The model defaults may change over time in the OpenNMT github and this would improve the ability for others to reproduce the work.

We understand the usefulness of such a list. We therefore added the configuration file generated by OpenNMT for pretraining to the Supporting Information, as the Supplementary Data 4. Furthermore, the exact commands used for training are available in the new GitHub repository at <https://github.com/rxn4chemistry/paragraph2actions>.

- Is the vocabulary size actually 16k or is that the maximum allowed? If it is the max allowed then do the sentences actually contain 16k where then the list is truncated, or less?

We understand that our statement about the vocabulary size could be misunderstood. We reformulated the paragraph to make it clear that its size is actually 16k.

SI:

- In Note 2 acknowledge that these actions cover most batch chemistry procedures but in the current form (which can easily be extended) does not capture some emerging chemistries such as electrochemistry (voltage, wattage, electrode type, etc..) and light mediated transformations (wavelength). The reviewer believes that these will be desirable in robotic execution for difficult to access structures.

We agree with the reviewer on this point and added a mention of such operations in Supplementary Note 5.

Other:

- There is a full reference and DOI for OpenNMT technical report (<https://www.aclweb.org/anthology/P17-4012/>) which should be used in place or in addition to ref 26

We replaced the reference 26 accordingly.

- When discussing datasets such as Reaxys, include Scifinder (CAS) since it is also a human curated dataset of comparable size and diversity.

We updated the manuscript and now mention SciFinder in the introduction.

- How does model handle “general procedures” where no specific reagent(s) are given. This is often seen in procedures where the same reaction is performed on multiple similar substrate. It seems the actions would still be translated but a short sentence acknowledging that this would not be useful for robotic execution without specific chemicals is warranted.

eg:

Specific Procedure: To a suspension of methyl 3-(7-amino-2-[(2,4-dichlorophenyl)(hydroxymethyl)]-1H-benzimidazol-1-yl)propanoate (6.00 g, 14.7 mmol) and acetic acid (7.4 mL) in methanol (147 mL) was added acetaldehyde (4.95 mL, 88.2 mmol) at 0° C...

General Procedure: To a suspension of the benzimidazole (6.00 g, 14.7 mmol) and acetic acid (7.4 mL) in methanol (147 mL) was added the aldehyde (4.95 mL, 88.2 mmol) at 0° C...

The model does not make a difference between specific and general procedures. It extracts compounds independently of whether they are specific ("methyl 3-7-amino-2-[(2,4-dichlorophenyl)(hydroxy)methyl]-1H-benzimidazol-1-ylpropanoate", etc.) or general ("aldehyde", etc.). For the task at hand, we believe that treating both equally is the correct thing to do – our goal is to extract actions from the experimental procedures, no more, no less. This goal is compatible with both general and specific procedures.

We added a comment in the manuscript to mention that both specific and general reagents are supported.

- The sentence “In the second experiment, the dataset is augmented as described above to produce 20000 samples (“augmented”)” the details of how the data is augmented are not clear or where above it is described.

In the first submission, the approach for data augmentation was described in the “Annotations” section. We moved it to its own section called “Data augmentation” for improved visibility and readability. We also added Table 6 to provide an example of data augmentation.

- It is recommended to open source the code so that it can be reproduced by researchers that have access to Pistachio. From the text, it does not appear that much of the code is proprietary (OpenNMT and sentencepiece are already open) so putting together the pipeline and publishing it on github should not be an obstacle.

As explained in response to Referee #1 and amended in the text, we open-sourced the definitions of the actions and the code to train the transformer (<https://github.com/rxn4chemistry/paragraph2actions>). The rule-based tools are IBM proprietary and are not available for distribution.

Reviewer #3 (Remarks to the Author):

In this paper the authors propose an approach for automatically extracting experimental procedures for organic chemical synthesis from English text, (specifically, the experimental procedures from patents). Their approach consists of annotating raw text sentences with corresponding sequences of synthesis actions and their parameters, then training a supervised neural network model, originally developed to perform the task of machine translation (automatically converting human-language sentences from one language to another), to output the correct sequence of actions.

This paper describes a solid contribution in the area of scientific information extraction / chemical informatics. Their main technical contribution is a manually annotated dataset of experimental procedures (1764 sentences), including an ontology for chemical synthesis steps and their parameters. They show that an ensemble of off-the-shelf machine translation models can be trained on this data to achieve while-sentence accuracy of 64.5% (i.e. 64.5% of sentence-level action sequences in the test set of 352 such sequences were exactly reproduced by the model).

They also suggest that pre-training the model with weak supervision from rule-based extractions is potentially useful in this setting, though this is not demonstrated in the experimental results, since no results are reported for a model lacking this pretraining, or comparing to a simpler baseline such as pretraining on the raw text with a self-supervised language modeling objective (e.g. BERT (Devlin et al. 2018)).

Training a model on the annotated data only (no pretraining) is a useful comparison, which we added to the paper (Results and Supplementary Note 2). Pretraining on the raw text with BERT would require considerable and non-trivial post-processing to make it compatible with the translation task of the present work, or to enable it to predict actions in any format. We therefore did not include BERT in the manuscript.

In addition to more/better baseline models, I would like to see a more detailed quantitative analysis of experimental results. Most reported analysis is qualitative, but it would be interesting to see model performance broken down by e.g. action label. This would be more feasible under a different evaluation paradigm (proposed in more detail below). I would also like to see a subset of examples annotated by different annotators in (at least) duplicate in order to report inter-annotator accuracy (e.g. Fleiss' kappa), as is standard, to measure the quality/consistency of the annotation. There are many cases where the mapping from text to actions is potentially ambiguous.

Indeed, breaking down the model performance by action label is a good idea. We added Table 5 and Figure 1 to illustrate this. In particular, in Table 5, one can see how many times the action label is predicted correctly, and what portion thereof also has correctly predicted action properties. Figure 1 intuitively illustrates the kinds of errors of the model, i.e. what action types are predicted instead of other ones.

We refer to the comments below for a discussion about the evaluation and about annotator accuracy/agreement.

Some of the most important limitations of this work are the annotation schema's inability to capture nonlinear experimentation, e.g. when a product is divided and subsequently used in two different experimental pathways, or connecting to an earlier described procedure. These

limitations are described in the paper but I would like to see more quantification justifying these decisions, such as the statement that “experimental procedures usually contain very few cross-sentence dependencies.” As it stands I’m concerned that the proposed data represent only a substantially simplified subset of the actual problem, and thus have limited potential for real-world impact.

We address this concern in detail further below. It turns out that the nonlinear procedures that cannot be captured by our approach represent a very small fraction of experimental procedures found in patents.

Another concern I have is regarding evaluation. Both BLEU score and whole-sentence accuracy compare the exact order of the series of experiment actions and parameters converted to text. This evaluation potentially unfairly favors the authors’ proposed technique, which is adapted from machine translation and is thus trained to produce the sequence of actions and their parameters converted to natural language text. However, a new system implementing a more structured approach might produce the same annotations as an ordered list of actions and (importantly, potentially unordered) parameters for each action. In order to be directly comparable to the proposed approach, this new system’s output would then have to be converted to text as well, potentially introducing errors in this superficial text conversion process that do not correspond to errors in the actual action sequence. For example, naive conversion might result in differently ordered parameters for an action, which unfairly favors models trained to explicitly produce action parameters in a certain order, despite this being an irrelevant artifact of the conversion to text. Indeed, a model that is trained to be invariant to these orderings might be more robust to variations in the text, but would potentially be penalized under BLEU or whole-sentence accuracy as proposed. An alternative approach that would resolve my concerns would align predicted actions with the gold-standard output to predict action accuracy (using e.g. Levenshtein distance), then for each correctly-aligned action compare order-invariant extracted parameters. BLEU is a poor approximation (by n-gram overlap) of this more exact evaluation that was designed for the much more ambiguous application of machine translation (where, unlike here, there may be many different correct translations), where it has also been shown to have issues such as poor correspondence to human acceptability judgements.

We thank the referee for his analysis and pertinent comments. In the following, we will explain why, in our view, both the BLEU score and the whole-sentence accuracy, without being perfect, are adequate measures.

Although both metrics rely on a text representation of the actions, they can just as well be used to measure the performance of a model not based on text representations (such as the rule-based model). It is true that the BLEU score (and the full-sentence accuracy) requires a predetermined ordering of the parameters for the actions. For a translation-oriented model, this ordering will be learnt by the machine-learning model, and therefore such models do not need an ordering-invariant. As noted by the referee, other (non-text-based) models may predict an unordered set of parameters. In such a case, the function to convert the action with its set of properties to a string representation (now available open-source) will automatically order the parameters correctly – which will produce a string that can be scored adequately with BLEU, also for such models. The conversion to a string representation is robust enough (see its open-source implementation).

The suggestion to include the Levenshtein distance is a very good one and we included it in the paper. Surprisingly, it does not correlate with the full-sentence accuracy as well as the BLEU score.

More detailed notes for the authors:

- Would like to know earlier in the text (abstract or introduction) what the neural network model is going to look like (encoder-decoder/ seq2seq). “transformer” is often used to just refer to the multi-head self-attention (versus the original encoder-decoder application of it).

We agree with the referee that readers could appreciate to have this information earlier in the text. In addition to explaining the model better in "Methods", the abstract and introduction now give more details about the model architecture (yet still in a concise form).

- Would be interesting to see discussion of information required for a human to successfully conduct an experiment versus a machine — certainly a human has a great deal of practical knowledge on experimentation that will not be explicitly stated in the text, resulting in extractions that are actionable for a human but far from actionable by a machine without an additional great deal of knowledge being encoded/curated from somewhere. How do you address this missing (implicit) information in your annotation scheme or model? For example, the text may simply say “dry”, implying the DryInVacuum operation since the product is a solid. “FollowOtherProcedure” does this to some extent. There is discussion of implicit actions, which is handled by a sentence with e.g. only one verb being mapped to many actions, or heat may go to “set temp” in some cases or “stir” in another. How does the model perform on these types of examples? Do you have annotation guidelines for handling these consistently? What is your inter-annotator agreement?

In this work, we solely focus on extracting the information that is present in the experimental procedures, no more than that. As such, we do catch some implicit information, as long as there is a hint of it in the text. For instance, “after filtration” will result in a Filter action, “after 10 minutes” in a Wait action, and “the aqueous layer ...” in a CollectLayer action (as explained in the manuscript). To take again the examples raised by the referee (“dry”, “set temperature”, “stir”), our experience is that these things can nearly always be deduced correctly from their context – it is possible to encode this in the rule-based model and annotate it accordingly for refinement, which is then sufficient for the ML model to also infer the correct action in these cases. What would not be caught at the moment is information that is difficult to determine from the context. For instance, if a compound is known to be air-sensitive, a chemist would know that the reaction must be done under inert conditions, while this will be missed by the machine unless a corresponding rule for such chemicals is formulated.

For the question regarding the annotation guidelines and inter-annotator agreement, we refer to our answer further below.

- How does XDL compare to the output of your model? Is substantial post-processing, addition of implicit information required to map from your annotations to XDL?

XDL describes procedures for synthesis in a robot-friendly fashion as well. It is formatted as XML and also allows for storing actions and associated properties. In this respect, it is more verbose than the string representations we define in our manuscript.

XDL is designed to be closer to the operations to execute on a given robotic system. When converting an experimental procedure to XDL, it will therefore also add apparatus information and other details implicitly, even details that are not present in the original experimental procedure.

Accordingly, the mapping of "our" actions to XDL would require the addition of those same values, in addition to the conversion from "our" action types to the XDL action types.

- How do you deal with branched synthesis procedures in your data? Are they simply ignored and not annotated?

Sentences corresponding to branched procedures correspond to InvalidActions. In the annotation set of 1764 sentences, this corresponds to four sentences (see answer further below for more details). These sentences are part of the dataset: it is desirable for the model to recognize such sentences and tag them accordingly.

We updated the manuscript text to clarify this.

- Breakdown by action types? e.g. MakeSolution seems hard. How often does InvalidAction happen, and is it included in your performance metrics? Is NoAction included in performance evaluation?

We thank the referee for the suggestion to break down the metrics by action types. We added this as Table 5. Both InvalidAction and NoAction are included in the performance evaluation – we consider it a correct prediction if the model considers an action to be invalid when the annotators considered a sentence to be unsupported. In Figures 3 and 4, one can now see the frequency of InvalidAction and NoAction.

- You report that a single annotator reviewed all the annotations, but it is standard to have multiple annotators annotate at least a subset of the data in order to report inter-annotator accuracy. As it stands we have no indication of annotator agreement, and thus consistency of annotations, for the dataset.

We thank the referee for his comments about the annotation process, especially about the inter-annotator agreement.

Let us consider the following points:

- Metrics:

To calculate the inter-annotator agreement, a metric for the distance between the annotations of two annotators is necessary. In the case of action sequences, it is not obvious what the best choice would be, since the annotations to compare are not simple categories, but rather lists of actions with associated values. This raises the question of how to evaluate annotations that differ by a) a missing action, b) a different action type, c) a different action property, etc.

We note that the strategy adopted for inter-annotator agreement in [dx.doi.org/10.1186/1758-2946-3-17](https://doi.org/10.1186/1758-2946-3-17) cannot be applied here since our annotations do not consist of the tagging / markup of text.

- Initial actions:

Before he/she starts annotating, the annotator is presented with an action sequence that was generated by the rule-based model (in the future: by earlier versions of the ML model). In other words, the annotators do not start from zero. There can therefore be a bias towards keeping the annotation similar to the one shown initially, which

would skew the inter-annotator agreement compared to annotations that start from zero.

- Iterative procedure:

The annotation process was, and still is, an iterative procedure. On the one hand, the definition of the actions (action types + associated properties) evolved iteratively. On the other hand, the sentences to annotate were not extracted randomly, but following diverse criteria that are explained in the paper. As a general trend, the sentences to annotate were more complex from annotation round to annotation round, which may well have an effect on the inter-annotator agreement (probably, the agreement would be higher for the easier sentences of early rounds).

- Guidelines:

A document with guidelines for annotation was available to the annotators to ensure consistency. We now include it in the revision of the paper as the Supplementary Data 3.

We agree that it would be nice to measure the inter-annotator agreement. Considering the above points, however, we did not (and still do not) think that it is straightforward to report a meaningful value for the inter-annotator agreement. Instead, we decided early on to review all annotations by a single annotator to make sure that the guidelines were followed correctly.

- I don't know if I agree that "experimental procedures usually contain very few cross-sentence dependencies." Could you quantify this? Perhaps in your limited framework where syntheses are linear, references to other parts of the text, and actions that depend on state are not really captured, cross-sentence dependencies that fit into your framework are rare. But I think it's quite common to see e.g. a product from a previous sentence referenced in a subsequent sentence, and making these connections would be very important in nonlinear experiments.

The experimental procedures present in our corpus describe single reaction steps. Therefore, it is rare that multiple reaction branches are described in a single procedure and rely on cross-sentence references. References such as "Compound A" or "compound of Example 1" are extracted by the model as compound names, as if the correct compound name was used directly. Also, references to previous sentences such as "the reaction mixture" or "the residue" are not problematic since they relate to the main reaction vessel anyway.

The problematic cases of non-linearity emerge when a sentence refers to a solution or compound that was generated several sentences earlier. To get an idea of the fraction of nonlinear experiments, we looked in our annotation dataset of 1764 sentences and found four such cases (for which the annotation is an InvalidAction):

- *"Then the reaction mixtures were combined and filtered."*
- *"The solution containing the product obtained previously is then added to this reaction medium very slowly, followed by addition of 23 mL (165 mmol) of triethylamine."*
- *"Separately, under nitrogen gas was in another vessel, a solution was prepared from methyl (S)-2-benzyloxycarbonylamino-3-phenylpropanoate (20.0 g, 63.9 mmol), dibromomethane (22.22 g, 127.8 mmol) and THF (40 mL) (liquor B)."*
- *"The residue and the above solid product were combined and triturated repeatedly with diisopropyl ether."*

This proportion corresponds to 0.23% of nonlinear sentences, and therefore we estimate that the fraction of nonlinear sentences is under 1%. This is consistent with what has been seen in the context of materials science (f.i., <http://arxiv.org/abs/1905.06939> mentions a value smaller than 1%).

We updated the manuscript to clarify this and now list the four sentences in the Supplementary Note 4.

- Why did you reduce the model size and remove label smoothing?

Tuning the hyperparameters for the pretraining step showed that larger models did not considerably increase the model performance, and that setting the label smoothing to 0 improved it. We also refer to the answer we gave above to a similar comment by Reviewer #1.

- Did you try just ensembling models with different random seeds, versus the augmented vs not augmented models? Could work just as well / better.

We thank the referee for this suggestion. For the revision of the manuscript, we ran, for each data augmentation approach, three experiments with different random number generator seeds. We list all of the results in Supplementary Note 2. Indeed, this works just as well as ensembling models with different data augmentation approaches.

- How did you decide when to stop pretraining (at 350k steps)?

As the performance on the pre-training validation set was sufficient at that point, we did not train the model further than that. For the updated model of the revision, we trained the model for 150k additional steps, which did not improve the performance on the pretraining dataset much.

- You may also want to cite: <https://www.aclweb.org/anthology/N18-2016/>

We thank the referee for making us aware of this work, which we now mention in the introduction.

Typos/grammar/style:

- Introduction:

- while scientists developed -> while scientists have developed

We corrected this.

- “Utilizing inherent rules”: not sure inherent is the word you want here — the rules are a heuristic attempting to get at some inherent properties specific to the data

We thank the referee for this comment. We replaced "inherent rules typical of each data item" by "rules specific to each data item".

- Important human effort -> substantial human effort

We corrected this.

- Not sure what “deep search” is

We rephrased the sentence to avoid using this term.

- Reviving use of robots, as in they were once used, then less so, and now mores anew?

We were referring to the increasing interest in robots in the context of synthesis. We reformulated this in the text.

- Results

- Would be great to see an example structured synthesis resulting from the example text at the beginning of results section. Ah! Table 2 should come

earlier, much closer to the example text.

We understand the reaction of the referee. We prefer not to include Table 2 before having defined the actions, and therefore we would like to keep Table 2 where it was. However, when introducing the example, we now refer to Table 2.

- You need to make it more clear what “augmentation” and “refined” mean in this table. I understand that refined is fine-tuning with manually annotated data, though that wasn’t completely obvious to me. Augmentation is not described until the methods section.

We improved this by mentioning data augmentation and not only in Methods, but in the Results section as well.

- Methods

- Figures 1 and 2: I don’t think you need an image (and especially not two) depicting the annotation framework; space could be better spent providing more quantitative analysis. This would be appropriate if the main contribution of your work were the annotation tool itself but I don’t think that’s the case. The annotation tool was an essential contribution to this work, since it enabled us to annotate sentences very efficiently. We believe that these figures provide a very good idea of the annotation process and carry some value for the reader. However, we agree with the referee that we don’t need two images to convey that message. We therefore combined both pictures into one.

REVIEWERS' COMMENTS:

Reviewer #1 (Remarks to the Author):

Thank you for better contextualizing this work by revising the introduction. There are still a few remaining clarifications that should be addressed in the text before the manuscript is ready for publication.

1. In response to my comment 8 about the "until ____" phrases, the authors state that storing extracted information as a raw string would miss the purpose of the current work since it would make post-analysis of that string necessary. However, most of the extracted action sequences (e.g., in Table 2) are designed to tag an action and associate it with a string field ("properties"). Much of the authors' original evaluation (e.g., using the BLEU score) is based on the generation of string data.
2. Comment 11 related to a claim about generality. There is still a paragraph in the text: "Although we focused on experimental procedures for organic chemistry extracted from patents, the approach presented in this work is more general. It can be adapted to any extraction of operations from text, possibly requiring new training data or the definition of new actions types to cover other domains adequately. For instance, our approach can be extended to other sources, such as experimental sections from scientific publications, as well as other fields, such as solid-state synthesis." This is still unsupported and should be removed. If the authors wish to claim that the model can "tackle generic documents" they should repeat their procedure on a variety of different texts, not just these structured patent paragraphs.
3. I do not understand the authors' clarification of 13b. For a rule based system, it still doesn't make sense why the word "Stir" wouldn't indicate a "STIR" action.
4. Were the 1764 sentences randomly selected from the corpus of 1M experimental procedures? The authors' response to my comment 14 only makes sense if the 1764 were randomly selected. It's a bit of a non-answer. The text should clearly state how these sentences were selected and provide at least a brief discussion of potential overlap in training and testing.
5. Perhaps the authors should refer to the OpenNMT documentation's statement of "The transformer model is very sensitive to hyperparameters" in the text.

Reviewer #2 (Remarks to the Author):

The authors have sufficiently addressed the concerns and the additions have made the manuscript much more accessible to a broad audience. Accept as is.

Reviewer #3 (Remarks to the Author):

I'm pleased overall with the authors responses and changes to the manuscript in response to my comments. I think the manuscript is quite improved. As in my original review I believe the paper describes a solid contribution in the area of scientific information extraction / chemical informatics, with the main contribution being the manually annotated data, and showing that a machine translation model trained on this data works reasonably well (at least, better than existing in-house methods). One of my main concerns with the approach was the inability of the annotation scheme to handle nonlinear experimental procedures, but the authors have done analysis to suggest that such nonlinear dependencies are very rare in this setting and data. They also included more empirical analysis in the results section following my suggestions, and released open-source code to the extent possible given that the work was completed at a company

including proprietary technology (models excluding the proprietary rules are also reported, though those using the proprietary information perform the best).

I would like to see the data made more easily available (without having to correspond with the authors). One common approach for text datasets that rely on licensed data (not to be redistributed) is to release code alongside a version of the dataset that does not include the licensed data, where the code can be run on a copy of the licensed data to obtain the full dataset. If not prohibitively difficult this might be a nice way to make the dataset more available to others, and thus accelerate research in this direction.

You used the pronoun "he" in your response to refer to the reviewer (me). In the future please don't assume the gender of your reviewer.

Small comments:

In Table 4, you may want to use different colors or annotation (e.g. bold, italic, underline, font) that is more friendly to the colorblind to indicate errors.

Point-by-point response to issues raised by the referees

Reviewer #1 (Remarks to the Author):

Thank you for better contextualizing this work by revising the introduction. There are still a few remaining clarifications that should be addressed in the text before the manuscript is ready for publication.

1. In response to my comment 8 about the “until ___” phrases, the authors state that storing extracted information as a raw string would miss the purpose of the current work since it would make post-analysis of that string necessary. However, most of the extracted action sequences (e.g., in Table 2) are designed to tag an action and associate it with a string field (“properties”). Much of the authors’ original evaluation (e.g., using the BLEU score) is based on the generation of string data.

It is correct that most other extracted properties are also strings. However, their context is quite specific: temperature, duration, compound name, etc. This makes a subsequent interpretation of the values straightforward, for instance for analysis or for application on a robotic system. The variability of what comes after "until" is very high and cannot automatically/easily be interpreted.

In the manuscript, we updated the comment about such sentences to clarify this.

2. Comment 11 related to a claim about generality. There is still a paragraph in the text: “Although we focused on experimental procedures for organic chemistry extracted from patents, the approach presented in this work is more general. It can be adapted to any extraction of operations from text, possibly requiring new training data or the definition of new actions types to cover other domains adequately. For instance, our approach can be extended to other sources, such as experimental sections from scientific publications, as well as other fields, such as solid-state synthesis.” This is still unsupported and should be removed. If the authors wish to claim that the model can “tackle generic documents” they should repeat their procedure on a variety of different texts, not just these structured patent paragraphs.

The paragraph quoted by the referee relates to the approach taken in our work, not to the models that we trained. Our approach is general, the trained model is not; we do not claim that the model can be used without change in those fields. For the initial revision, we took into account the original comment 11 and we specified (see quoted text): "possibly requiring new training data or the definition of new action types".

To clarify this further, the sentence mentioning other fields such as solid-state synthesis now emphasizes that the data and action definitions will need adequate changes for the approach to be applied there.

3. I do not understand the authors’ clarification of 13b. For a rule based system, it still doesn’t make sense why the word “Stir” wouldn’t indicate a “STIR” action.

As explained in the manuscript, the actions are determined not only by recognizing words, but also their context and role in the sentence. With the rule-based model, the word "stir"

leads to a STIR action only if it is the central verb of the (sub-)sentence. This is not the case in the example 13b, and therefore the sentence is not converted to a STIR action. This is a deliberate decision in the design of the rule-based model: the context of the action words is essential in most cases. As an example, "stir" may also refer to a noun with a different meaning than stirring in a chemical sense.

4. Were the 1764 sentences randomly selected from the corpus of 1M experimental procedures? The authors' response to my comment 14 only makes sense if the 1764 were randomly selected. It's a bit of a non-answer. The text should clearly state how these sentences were selected and provide at least a brief discussion of potential overlap in training and testing.

The sentences were selected following specific criteria (see Methods, Annotations in the manuscript). The selection is therefore not random, although the experimental procedures in which to look for such sentences were picked at random. The bias in the selection is visible in Fig 3, left (in the revised manuscript: Figure 2c).

We updated the paragraph describing the selection of the samples to clarify this. We now also provide the script for the selection of annotation samples in the GitHub repository. After the details of splitting the dataset in Methods, we also added a paragraph discussing the potential overlap of training and testing data.

5. Perhaps the authors should refer to the OpenNMT documentation's statement of "The transformer model is very sensitive to hyperparameters" in the text.

We added this indication to the manuscript.

Reviewer #2 (Remarks to the Author):

The authors have sufficiently addressed the concerns and the additions have made the manuscript much more accessible to a broad audience. Accept as is.

Reviewer #3 (Remarks to the Author):

I'm pleased overall with the authors responses and changes to the manuscript in response to my comments. I think the manuscript is quite improved. As in my original review I believe the paper describes a solid contribution in the area of scientific information extraction / chemical informatics, with the main contribution being the manually annotated data, and showing that a machine translation model trained on this data works reasonably well (at least, better than existing in-house methods). One of my main concerns with the approach was the inability of the annotation scheme to handle nonlinear experimental procedures, but the authors have done analysis to suggest that such nonlinear dependencies are very rare in this setting and data. They also included more empirical analysis in the results section following my suggestions, and released open-source code to the extent possible given that the work was completed at a company including proprietary technology (models

excluding the proprietary rules are also reported, though those using the proprietary information perform the best).

I would like to see the data made more easily available (without having to correspond with the authors). One common approach for text datasets that rely on licensed data (not to be redistributed) is to release code alongside a version of the dataset that does not include the licensed data, where the code can be run on a copy of the licensed data to obtain the full dataset. If not prohibitively difficult this might be a nice way to make the dataset more available to others, and thus accelerate research in this direction.

As we clarified in our reply, the only data that remains available upon request are the experimental procedure sentences extracted from Pistachio (owned by NextMove Software). All the other data is available without contacting us. Including source code available already now from GitHub.

Pistachio is a proprietary (NextMove Software) DB. The distribution of NextMove Software data (similar to NPG data) is legally and contractually regulated by the terms and conditions that come with the purchase of NextMove Software DB. We are not in a position to distribute freely this data. Still, upon request, we can verify with NextMove Software if the person holds subscription rights to access the portion of NextMove Software DB for which we determined action sequences (hand-annotated and with the rule-based model) and provide access. We appreciate the comment of the reviewer but there are legal bindings that each of us, including the reviewer, needs to comply with.

You used the pronoun “he” in your response to refer to the reviewer (me). In the future please don’t assume the gender of your reviewer.

We apologize for the oversight.

Small comments:

In Table 4, you may want to use different colors or annotation (e.g. bold, italic, underline, font) that is more friendly to the colorblind to indicate errors.

We followed this suggestion and removed the colors in Table 4.